



# Semi-equilibrated global sea-level change projections for the next 10,000 years

Jonas Van Breedam[1], Heiko Goelzer[1][*], Philippe Huybrechts[1]

[1] Earth System Science and Departement Geografie, Vrije Universiteit Brussel, Pleinlaan 2, B-1050 Brussel, Belgium
[*] Now at: Institute for Marine and Atmospheric research Utrecht, Utrecht University, The Netherlands

*Correspondence to*: Jonas Van Breedam (jonas.van.breedam@vub.be)



**Abstract.** The emphasis for informing policy makers on future sea-level rise has been on projections by the end of the 21$^{st}$ century. However, due to the long lifetime of atmospheric $CO_2$, the thermal inertia of the climate system and the slow equilibration of the ice sheets, global sea level will continue to rise on a multi-millennial timescale even when anthropogenic $CO_2$ emissions cease completely during the coming decades to centuries. Here we present global sea-level change projections due to melting of land ice combined with steric sea effects during the next 10,000 years calculated in a fully interactive way with the Earth System Model of Intermediate Complexity LOVECLIMv1.3. The climate forcing is based on the Extended Concentration Pathways defined until 2300 AD with no carbon dioxide emissions thereafter and the inclusion of a methane-emission feedback for the highest forcing scenario, equivalent to a cumulative $CO_2$ release of around 460 to 5800 GtC. After 10,000 years, the sea-level change rate drops below 0.05 m per century and a semi-equilibrated state is reached. The Greenland ice sheet is found to nearly disappear for all forcing scenarios. The Antarctic ice sheet contributes only about 1.6 m to sea level for the lowest forcing scenario with a limited retreat of the grounding line in West Antarctica. For the higher forcing scenarios, the marine basins of the East Antarctic ice sheet also become ice free, resulting in a sea-level rise of up to 27 m. The global mean sea-level change after 10,000 years ranges from 9.2 m to more than 37 m. The projections of multi-millennial semi-equilibrated sea-level rise for a given $CO_2$ forcing are shown to be in good agreement with geological archives.

# 1 Introduction

Modern sea-level rise started at the end of the 1800s and has accelerated over the course of the 20$^{th}$ century (Church and White, 2011, Hay et al., 2015) with an unprecedented rate over the observational era during the first part of the 21$^{st}$ century (Watson et al., 2015). The rate of sea-level rise was 2.5 times larger during the last decade than during most of the 20$^{th}$ century (Oppenheimer et al., 2019). So far, the horizon for most global mean sea-level change projections has been the end of the 21$^{st}$ century (Jackson and Jevrejeva, 2016, Kopp et al., 2017, Goelzer et al., 2020b, Seroussi et al., 2020). Sea-level change projections beyond 2300 are scarce (Clark et al., 2016). However, due to the long lifetime of carbon dioxide in the atmosphere (Archer et al., 2009b) and the thermal inertia of the climate system (Solomon et al., 2009, Gillett et al., 2011, Goelzer et al., 2012), sea level is expected to continue to rise on a multi-centennial to multi-millennial timescale. Moreover, the large ice sheets themselves have a very long response time to any perturbations (Golledge, 2020) with the longest response time ranging from 10,000 years for accumulation rate changes to up to 100,000 years for temperature changes for the Antarctic ice sheet (Alley and Whillans, 1984).

On a multi-millennial timescale, changes in land ice volume (mass contribution) together with ocean density changes from thermal expansion (thermosteric contribution) and haline contraction (halosteric contribution) are the main components contributing to global sea-level change (Miller et al., 2005). The mass contribution comes from melting or growing of the Antarctic ice sheet, the Greenland ice sheet and glaciers and ice caps. The steric contribution is the expansion of ocean water



when it gets warmer and less saline or conversely, the contraction when water is cooling and gets more salty (Feistel, 2010). The magnitude of thermal expansion depends on the climatic temperature forcing and on the rate of oceanic heat uptake. Because of the large mass and slow turnover time of the deep ocean, the oceanic heat content (and hence thermal expansion)

will continue to rise in the ocean for centuries to millennia, even when surface warming has halted. Haline contraction – or expansion of ocean water due to freshening – is a smaller, though not negligible component of steric sea-level rise. Physically-based models have been used to study the different components contributing to global sea level on a multi-millennial timescale. For example, long-term ice sheet evolution and resulting sea-level change projections have been made for the next few millennia for the Greenland ice sheet (Charbit et al., 2008, Robinson et al., 2012, Applegate et al., 2015), the Antarctic ice

sheet (Huybrechts, 1993, Golledge et al., 2015, Pollard et al., 2015, Winkelmann et al., 2015, DeConto and Pollard, 2016) or both (Vizcaíno et al., 2008, Huybrechts et al., 2011).

Existing long term sea-level change projections including both the changes in land ice volume and the steric components extend for the next 2000 years (Levermann et al., 2013) or 10,000 years (Clark et al., 2016). These studies did not take the full coupling between the ocean and the atmosphere into account, which becomes important when the ice sheets lose large volumes

of freshwater. Goelzer et al. (2012) studied the committed sea level rise at the end of 3000 AD with a coupled model approach for a range of idealised $CO_2$ scenarios. So far, the study of semi-equilibrated sea level changes including all components of future sea-level change with the incorporation of feedbacks between the climate components has not been performed because of the high computational cost. The adopted trade-off between model complexity and model interactions on a multi-millennial timescale consists of the use of the Earth System Model of Intermediate Complexity LOVECLIM with a lower resolution but

fully integrated coupling.

In this study, we project the global mean sea level changes for the next 10,000 years in a fully interactive way between all major components in the climate system. The duration of the simulations allows the climate system to reach a semi-equilibrated state, long time after the cessation of anthropogenic greenhouse gas emissions. The greenhouse gas forcing follows the Extended Concentration Pathway (ECP) scenarios to span the likely range in climate uncertainty with the inclusion of a

methane-emission feedback in a warming climate.

## 2 Model description and initialisation

High resolution general circulation models (GCM) are the best tools to project climate changes until the end of the century or a few hundred years after that, but they are computationally too expensive to make millennial to multi-millennial projections. Earth System Models of Intermediate Complexitiy (EMIC) have a lower resolution and therefore allow for longer simulations

such as simulating the climate over the last millennium (Eby et al., 2013) or exploring climate-carbon cycle feedbacks during the next 1000 years (Zickfeld et al., 2013). Here we make use of the EMIC LOVECLIMv1.3 (Goosse et al., 2010) for the projections of global sea-level change on a multi-millenial timescale. LOVECLIM is one of the few EMICs with an ice sheet



component (AGISM) that is fully coupled to the other components of the climate system (ECBilt for the atmosphere – CLIO for the ocean and sea-ice – VECODE for the terrestrial biosphere), allowing for multi-millennial projections of sea-level

change. ECBilt is a quasi-geostrophic atmospheric model with truncation T21, corresponding to a resolution in longitude and latitude of approximately 5.625˚ and three vertical levels (Opsteegh et al., 1998). CLIO is a global free surface ocean GCM coupled to a thermodynamic sea-ice model, with a resolution of 3˚ in longitude and latitude and 20 unevenly spaced vertical levels (Goosse and Fichefet, 1999). AGISM, the ice sheet model component consists of 3-D thermomechanically coupled ice sheet models for the Greenland ice sheet and the Antarctic ice sheet (Huybrechts and de Wolde, 1999) and is run at a resolution

of 10 km for the Greenland ice sheet and 20 km for the Antarctic ice sheet. The relative coarse resolution is necessary to allow for long integrations (10,000 model years take 20+ days of computational time). The global glacier melt algorithm is based on the model of Raper and Braithwaite (2006) that considers a total sea-level equivalent (SLE) of 0.241 m (mountain glaciers contain 41% of this estimate while the ice caps are responsible for the other 59% of SLR potential). This is at the lower end of the range of total recent estimates of ice volume (0.24 – 0.40 m SLE) stored in all ~215 000 glaciers on Earth (Farinotti et al.,

2019), partly explained by the fact that the glacier model excludes glaciers in the periphery of the Greenland and Antarctic ice sheet. The steric sea level change component, thermal expansion and haline contraction, is calculated based on regional changes in ocean temperature and salinity. The ocean carbon cycle is not activated and greenhouse gas concentrations are prescribed (see Sect 3. and Appendix A).

The model has been used previously for ice sheet and climate change projections for periods in the past (Loutre et al., 2014,

Goelzer et al. 2016a, 2016b) and the future (Huybrechts et al., 2011, Goelzer et al., 2012). The version used here differs from LOVECLIMv1.3 for the Antarctic ice sheet component by including surface ablation on the ice shelves, an important process under future atmospheric warming. A tundra warming feedback is also included for retreating ice sheets by making a distinction between snow/ice albedo and tundra albedo. The snow/ice albedo is replaced by tundra albedo (with a lower albedo) once the ice sheet margin is retreating on land. In order to better represent the climate forcing for the ice sheet models, temperature and

precipitation anomalies are applied to eliminate the bias from a coarse resolution atmospheric model. The surface mass balance is calculated using the PDD method for both ice sheet models. The Antarctic ice sheet model uses the shallow ice approximation (SIA) for grounded ice and the shallow shelf approximation (SSA) for the ice shelves coupled across a one grid cell wide transition zone. Basal melting is calculated as a function of the mean oceanic heat input into the ice shelf cavities. Sea-level changes resulting from melting of the marine-based parts of the Antarctic ice sheet are corrected for bedrock

elevation changes (Goelzer et al., 2020a). Both ice sheet models have recently participated in the ISMIP6 initiative (Goelzer et al., 2020b; Seroussi et al., 2020).

Since many physical parameters in a climate model are uncertain, LOVECLIM is used to explore the climate response for a large combination of climate model parameter sets (Loutre et al., 2011). Here we have evaluated 4 different model parameter sets by testing the sea-level contribution for the period 1900-2300 AD. The 4 model parameter sets (P11, P22, P32a, P32b; see

Table S3) differ in their sensitivity to the applied greenhouse forcing (P32b > P32a > P22 > P11) and freshwater forcing in the





North Atlantic (P32a, P32b and P22 > P11). All climate model parameter sets yield climate simulations in agreement with observations over the past 500 years (Loutre et al., 2011). Model parameter set P22 is chosen for the multi-millennial integrations because of its mid-range contribution to sea-level at 2100 AD and 2300 AD in comparison with recent studies (Pörtner et al., 2019; Calov et al., 2018; Bulthuis et al., 2019; Tables S1-S2 and Figure S1). The mean annual temperature

anomalies over the ice sheets for 2070-2100 relative to 1970-2005 (+4.6˚C over Greenland and +3.8 ˚C over Antarctica for RCP8.5) correspond well with the mid to upper ranges of AOGCM projections over the polar regions (Fettweis et al., 2013, Barthel et al., 2019). With an equilibrium climate sensitivity of 2.3˚C, the model parameter set is at the lower end of IPCC AR5 estimates (likely range of 1.5˚C - 4.5˚C) due to a cold bias in the tropics.

The Greenland ice sheet and Antarctic ice sheet are spun-up from standalone experiments forced with reconstructed

temperature forcing above the ice sheet following ice core records for respectively two and four glacial-interglacial cycles to carry their long-term ice sheet history (Huybrechts et al., 2011; Goelzer et al., 2012). A quasi-equilibrium experiment is run with the aim to get a climate in equilibrium at 1500 AD with the ice sheet forcing. The ice sheet models in this experiment evolve very slowly without seeing the changes of the climate model. The climate model on the other hand is subject to changes in the ice sheet models (changes in freshwater flux, albedo and height changes) and equilibrates to these boundary conditions.

At the end of the quasi-equilibrium run (after 2500 years), a steady state is reached. A control experiment with constant pre-industrial greenhouse gas forcing showed that the drift in climate and ice sheet components is negligible. The climate information present at the end of the quasi-equilibrium experiment is used as input for the initialisation of the fully coupled experiments starting at 1500 AD.

The fully coupled model is run from 1500 to 2000 AD. LOVECLIM is forced during this historical period with known greenhouse gas concentrations, total solar irradiance and volcanic forcing. Now, all components of LOVECLIM interact dynamically and are run as an ensemble of five members. The ensemble members are identical except for small perturbations in the initial conditions. Five iterations of the reference state for the ice sheet models are performed to see if there is convergence for the climatic information (e.g. Greenland and Antarctic temperature forcing, oceanic heat content and

meridional overturning circulation variability). The experiment performing best at representing the climate over the last 500 years is chosen as input for the long-term future projections.

## 3 Scenario description

Six different forcing scenarios are constructed to span a large range of future greenhouse gas forcing. The main difference between the different scenarios is the atmospheric carbon dioxide forcing. Four of the $CO_2$ forcing scenarios are extensions of

the ECP scenarios (Meinshausen et al., 2011) and will be referred to as Multi-Millennial Concentration Pathways (MMCPs) with the same designation as the RCP scenarios (MMCP2.6, MMCP4.5, MMCP6.0 and MMCP8.5). The scenarios range from a peak in the $CO_2$ concentration of 443 ppmv in 2053 AD with temporarily negative emissions thereafter (MMCP2.6) to a



peak of 1962 ppmv in 2250 AD (MMCP8.5). Total recoverable carbon (oil, gas, lignite and coal) reserves (the total exploitable carbon) and resources (total carbon mass stored on earth, including the non-exploitable carbon) estimates are disputed, ranging

from < 1500 GtC reserves and 4000 GtC resources (Hasselmann et al., 2003) up to 2900 GtC reserves and 11000 GtC resources (McGlade and Ekins, 2015). In this study, the MMCP scenarios (extended ECP scenarios with zero emissions after 2300 AD) are equivalent to a total cumulative emission ranging from less than 200 GtC for scenario MMCP2.6 to more than 5000 GtC for scenario MMCP8.5 relative to the year 2000 AD (see Appendix: Table A3). This is comparable to the study from Clark et al. (2016) with an emission pulse of 1280-5120 GtC, but well below the study from Winkelmann et al. (2015) that assumes a

total carbon release of up to 10,000 GtC.

Two additional scenarios are constructed. The first one assumes that $CO_2$ concentrations follow ECP8.5 but emissions will cease completely by the year 2150 AD. This scenario is referred to as MMCP-break and is an intermediate forcing scenario between MMCP6.0 and MMCP 8.5. The second additional scenario (MMCP-feedback) assumes that methane emissions increase as a feedback to the warming climate (Table A3). The size of the methane reservoir is a major unknown and estimates

range from 500-3000 GtC (Piñero et al., 2013) to 5000-20,000 GtC (Dickens, 2011). Ruppel and Kessler (2017) made an extensive review to estimate the size of the methane hydrate reservoir and finds a converging value of around 2000 GtC. MMCP-feedback assumes a moderate methane release from methane hydrates of 600 GtC by adding constantly $CO_2$ after 2250 AD (from the peak concentration onwards) until the end of the simulations, in accordance with the experiments of Archer et al (2009a). At least 50% of the methane released from the hydrates would be anaerobically oxidised inside the seafloor and an

additional part is converted into $CO_2$ in the water column by aerobic oxidation (Treude et al., 2003). In our experiments, it is assumed that the methane released from methane hydrates is completely oxidised in $CO_2$ when it reaches the atmosphere.

The carbon dioxide concentrations follow the ECP scenarios until 2300 AD with zero emissions thereafter. Impulse response functions are used to construct the falloff of carbon dioxide concentrations (Appendix A). Carbon cycle models show that a perturbation of $CO_2$ can remain for tens of thousand of years in the atmosphere (Archer et al., 2009b, Lord et al., 2016). After

a perturbation, atmospheric carbon is taken up by the land biosphere (~100 years), ocean invasion (10-1000 years), $CaCO_3$ weathering (1000-10,000 years) and silicate weathering (10,000-1,000,000 years) to eventually restore the initial concentrations (Ciais et al., 2013). The more carbon is emitted, the slower the uptake of carbon dioxide in the ocean due to the reduction in buffering capacity (Archer et al., 2009b). The constructed carbon dioxide concentrations for the next 10,000 years take these effects into account in a schematic way and are shown in Figure 1a.

Methane (average lifetime of 9 years) and nitrous oxide (average lifetime of 131 years) concentrations are kept constant at their concentration in 2300 AD (Meinshausen et al., 2011) because of the high rates of natural emissions and the increase in natural emissions for a warming climate. The increase in natural emission of methane are ascribed to the increase in natural wetland methane emissions in a warmer climate (Kirschke et al., 2013). Higher decomposition rates in a warming climate are responsible for an increase in nitrous oxide emissions (Griffis et al., 2017). Other greenhouse gases (methane and nitrous





oxide) follow the trajectory until 2300 AD as presented in Meinshausen et al. (2011) and either decrease to zero when the trend is declining and the lifetime of the gases is short (chlorofluorocarbons, $CCl_4$ and $CH_3CCl_3$) or stay constant over the next 10,000 years when the overall trend is unclear (hydrofluorocarbons and hydrochlorofluorocarbons).

The orbital parameter variations are included in the climate forcing. However, the insolation differences in the polar regions are rather small (summer insolation will slightly increase in the northern hemisphere and decrease in the southern hemisphere

compared to the present-day) on account of a low eccentricity and therefore its influence on the radiative forcing is limited. The future solar forcing is unknown and therefore the last 11-year solar cycle is repeated for the next 10,000 years.

## 4 Climate response and global sea-level budget of individual terms

In the following sections, the climate response to the greenhouse gas forcing and insolation variations and the sea-level changes resulting from mass changes of the Greenland ice sheet (GrIS), the Antarctic ice sheet (AIS) and glaciers and ice caps, and

thermal expansion and haline contraction of the ocean water are shown. The relative importance of each contribution to global mean sea level (GMSL) following a specific scenario is discussed.

### 4.1 Climate response

Global mean surface air temperature (SAT) is projected to increase between 1.7 ˚C and 5 ˚C following the strong increase in greenhouse gas forcing (Figure 1). In the polar regions, the reduction in sea ice area and its large effect on absorption of heat

through the ocean surface albedo leads to an amplified response to the climatic forcing. Moreover, the height-mass balance feedback and the albedo-temperature feedback causes the GrIS mean annual SAT to rise even further as ice is melting. This is most obvious for the mean annual SAT anomaly using scenarios MMCP2.6 to MMCP-break where the SAT is increasing strongly when the GrIS melted an equivalent of around 2 m sea level (Figure 1c). The increase in Antarctic mean temperature over the next 10,000 years shows a very large spread between the different scenarios. This is partly caused by a decreasing

sea-ice area trend and its influence on the albedo, where the retreat will be larger for the high forcing scenarios (Figure 9). Snow accumulation is projected to increase for all forcing scenarios due to the increase in atmospheric water vapor, whereas the increase in accumulation will be larger for the strongest warming scenarios (up to 16 % for scenario MMCP-feedback over the Antarctic continent). Freshwater fluxes from the Greenland and Antarctic ice sheets lead to temporarily reduced temperatures and a delayed peak warming over the ice sheets.

### 4.2 Thermal expansion and haline contraction

The projections for thermal expansion and haline contraction show an increase following global mean temperature perturbations with the longer response time for the higher forcing scenarios. Thermal expansion and haline contraction rise rapidly before showing a period of stabilization for all scenarios. Except for scenario MMCP2.6 and MMCP4.5, this period is



followed by another more gradual increase in sea-level rise during the second half of the simulations (Figure 4). This is most

pronounced for scenario MMCP6.0 and MMCP-break where the steric contribution to GMSL rise reaches respectively 0.3 and

0.4 m between 7000 AD and 12,000 AD. Oceanic temperatures have already equilibrated to the forcing and at 12,000 AD,

deep ocean temperatures are slightly lower than temperatures at 7000 AD, following the slow decrease in global mean

temperature anomalies (Figure 11). Therefore, we attribute the sea-level rise during the second half of the simulations to the

continued freshwater release from the Antarctic ice sheet. On the other hand, the steric contribution to GMSL slightly decreases

for scenarios MMCP2.6 and MMCP4.5 after a peak at 3000 AD. Freshwater fluxes are still high due to a slow melting of the

GrIS, suggesting that the lower increase in global mean temperatures decreases the response time. At the end of the simulations,

the total range of the steric component to GMSL change is between 0.4 m (MMCP2.6) and 3 m (MMCP-feedback) .

### 4.3 The Greenland ice sheet

The GrIS volume changes because of an imbalance between the surface mass balance (SMB), the difference between

accumulation and ablation, and iceberg calving at the ice sheet margin. Surface runoff is projected to increase for any of the

forcing scenarios, though the magnitude differs significantly. After 4000 years, surface runoff exceeds accumulation and the

surface mass balance becomes negative even for the lowest forcing scenario (Figure 2c). In a high warming scenario, this

might happen already at the end of the 21$^{st}$ century (Figure 2d). When looking at all the mass balance components, one can see

that the relative importance of iceberg calving decreases when the ice sheet retreats on land. The large amounts of meltwater

result in a total freshwater flux anomaly of 0.03 to 0.05 Sv for the three higher forcing scenarios and is sustained for 1000 to

1500 years (Figure 7a). The Atlantic Meridional Overturning Circulation (AMOC) almost completely shuts down for the two

highest forcing scenarios (Figure 8). This results in a local cooling south of the GrIS and a delayed peak warming above the

Greenland ice sheet (most pronounced for MMCP-feedback; Figure 1c), pointing to the importance of simulations with a

coupled ocean-atmosphere-ice sheet model. The AMOC recovers again after the ice sheet has melted entirely and even

becomes stronger for the higher forcing scenarios (Figure 10).

### 4.4 The Antarctic ice sheet

The Antarctic continent is very cold and there is almost no surface melt over the grounded ice at present. There is a very

distinct response of the AIS to a low or a high forcing in terms of mass loss. Looking at the two extremes, ice discharge at the

grounding line is projected to increase slightly for scenario MMCP2.6, similar to previous suggestions that increased ice

discharge is a response to increased accumulation (Winkelmann et al., 2012). The ice shelves thin for 1700 years because of

atmospheric warming, while basal melting rates below the ice shelves remain low with a mean value of 0.3 m per year over

all the ice shelves. The ice shelves lose 2/3 of their volume, but they recover again to reach their initial volume after 5000

years when surface ablation becomes negligible In scenario MMCP-feedback, even the large ice shelves around Antarctica

disappear nearly completely after 300 years as a combination of atmospheric warming (and subsequent surface ablation) and



a quadrupling in ice shelf basal melt rates. Ice discharge across the grounding line is projected to increase for about 500 years and drops below present-day values after 1700 years when the West Antarctic ice sheet has collapsed and the East Antarctic ice sheet starts to retreat land inwards. The increase in surface runoff is even stronger and exceeds accumulation at the end of the first millennium for about 3000 years, after which surface runoff and ice discharge along the grounding line stabilize and account each for about half of the mass loss (Figure 3d). The freshwater input in the Southern Ocean increases with 0.14 Sv

for the highest forcing scenario and remains elevated until the end of the simulations with an anomaly of 0.07 Sv due to continued ice sheet melting and increased runoff over land (Figure 7b). As a consequence of the freshwater release into the Southern Ocean, the strength of Antarctic Bottom Water (AABW) declines with 23 % for MMCP2.6 and up to 77 % for MMCP-feedback during the next 500 years due to rapid ice sheet melting (Figure 8). These freshwater fluxes from the AIS lead to reduced oceanic warming in the vicinity of Antarctica and relatively low basal melt rates in our experiments (mean

basal melt rates of 0.5-1.1 m per year for scenario MMCP-feedback during the coming centuries). Mean SAT anomalies above the Antarctic ice sheet reach a maximum between 5000 and 6000 AD when the ice sheet retreats on land and the albedo-temperature feedback sets in (Figure 1d). The mass balance components stabilize during the second half of the simulations with iceberg calving and ablation both accounting for half of the mass loss, a situation comparable to the Greenland ice sheet today.


There is a very large difference in ice sheet geometry of the AIS after 10,000 years for the lowest and highest forcing scenario. In scenario MMCP2.6, the grounding line retreats mostly in the Weddell Sea and the Ross Sea basin, with almost no retreat for the East Antarctic ice sheet (Figure 3a). For the high emission pathway, the West Antarctic ice sheet collapses and grounding line retreat initiates in the Wilkes subglacial basin (Figure 3b). Due to isostatic rebound of the Earth's crust, more

land is exposed above sea-level at 12,000 AD compared to the situation at 7000 AD and the ice sheet margin becomes land-based for most of East Antarctica as has been proposed for warm intervals during the mid-Miocene (Gasson et al., 2016; Frigola et al., 2018).

### 4.5 Glaciers and ice caps

Glaciers and ice caps have a decadal to century timescale response and are vulnerable to small perturbations in the climate

forcing. For the highest forcing scenarios MMCP-break, MMCP8.5 and MMCP-feedback, the glaciers and ice caps melt away entirely in the coming 1000 to 2000 years. Under scenario MMCP2.6 it takes up to 3000 years before the glaciers and ice caps lose most of their ice volume. On the multi-millennial timescale, the differences in the glaciers and ice caps contribution to sea-level change are very small compared to the other components with total contribution between 0.23 and 0.24 m for any of the forcing scenarios.



## 5. Global mean sea-level change

Total GMSL changes in our experiments range between 9.2 m and 37.4 m at the end of the 10,000 year experiments (Figure 5a). Moreover, the rates of GMSL rise vary substantially between the different scenarios and over time. We investigate the relation between cumulative $CO_2$ emissions and GMSL change, somewhat similar to the notion of a relationship between global warming and cumulative $CO_2$ emissions by a certain time, expressed as the Transient Climate Response to cumulative carbon Emissions (TCRE: Matthews et al., 2018) or the Multi-millennial Climate Response to cumulative carbon Emissions (MCRE: Frölicher and Paynter, 2015). In Fig. 5b, the realised GMSL rise after each 1000 years is shown as a function of the total cumulative $CO_2$ emission after 2000 AD for each experiment.

In case total cumulative $CO_2$ emission exceed 2000 GtC, GMSL change rates are highest during the first 2 millennia. However, when cumulative $CO_2$ emissions stay below 200 GtC (MMCP2.6), the peak rates in GMSL change occur only 4000 to 8000 years from now due to slow melting of the GrIS and ensuing feedbacks. A total sea level rise of 8.8 m is achieved after 10,000 years , even though $CO_2$ emissions already ceased before the mid 21$^{st}$ century. For the other experiments, the rate of sea-level change decreases with time. All experiments approach a semi equilibrium after 10,000 years (Figure 5a). The rate of global sea-level change is then still positive for all simulations, albeit reduced to about 0.05 m per century for the experiment with the largest cumulative $CO_2$ emission (MMCP-feedback). This reflects the long equilibration time for the ice sheets and the ocean within the coupled climate system to adapt to temperature changes.

In our simulations, Greenland is the largest contributor to GMSL rise up to a cumulative $CO_2$ emission of 2000 GtC. The grounding line of the West Antarctic ice sheet (WAIS) retreats several 100 km inland using scenario MMCP2.6 and MMCP4.5. The WAIS disintegrates completely for cumulative $CO_2$ emissions around 2000 GtC, but East Antarctica still remains mostly unaffected. For higher cumulative emissions, marine-based parts of East Antarctica start to disintegrate and Antarctica becomes the largest source to GMSL rise.

The total sea-level rise after 10,000 years includes the (nearly) entire melt of the Greenland ice sheet as well as glaciers and ice caps. The spread is largest for the Antarctic contribution taken over all scenarios, ranging from 1.6 to 27 m (corresponding to ~200 to ~5000 GtC cumulative $CO_2$ emissions after 2000 AD). The contribution from thermal expansion and haline contraction ranges between 0.4 and 3 m. Table 1 gives an overview of the sea-level contribution and the relative share to GMSL of each component for all forcing scenarios.

## 6. Long-term sea level rise in the light of the geological record

An interesting test for our sea-level change projections can be provided by the geological record of sea-level high stands as a function of the inferred atmospheric carbon dioxide concentration. Estimates of paleo sea level variations  range from a lowstand of -120 m for $CO_2$ concentrations around 180 ppmv to a highstand of +65 m for $CO_2$ concentrations up to 1200 ppmv





(Alley et al., 2005, Foster and Rohling, 2013). Such data suggest a linear sigmoidal behaviour for (semi)-equilibrated periods in the past, where sea-level high stands changes abruptly for deviations of the greenhouse gas forcing from the pre-industrial due to the build-up of the large northern hemisphere ice sheets during glacial periods and the total melting of the Greenland and Antarctic ice sheet for a high $CO_2$ greenhouse world during the Eocene. Assuming that our climate (and sea-level change) components are nearly equilibrated after 10,000 years, we compare our GMSL changes with the geological archive (Foster

and Rohling, 2013). For this comparison, it is important to realise that the time scale of carbon input in the atmosphere may be a critical parameter as the peak atmospheric $CO_2$ concentration shows a strong dependence on the emission pathway. However, simulations with biogeochemical models show that the mean atmospheric $CO_2$ concentration over multiple kyr is mostly independent from the duration of carbon release (Zeebe and Zachos, 2013). Moreover, on a multi-centennial timescale, global mean temperature perturbations converge to a single value, suggesting a pathway independence of cumulative emissions

(Zickfeld et al., 2012, Rogelj et al., 2016). We therefore opt to average the $CO_2$ concentration over the next 10,000 years, suggesting that the mean concentration represents the multi-millennial temperature change at best.

Figure 6 shows the GMSL change after 10,000 years as a function of the mean atmospheric $CO_2$ compared with the geological archive. We compared a piecewise linear fit of the data obtained in this paper (red line) with the best fit of the geological data (blue line, polynomial with exponent 2; data from Foster and Rohling, 2013). Both of the fitted lines compare quite well. There

is a somewhat larger discrepancy for the highest $CO_2$ concentration, but the (near) total melting of all ice on Earth for a mean $CO_2$ concentration of 1200 ppmv is based on only a few geological data points. Moreover, a further upward GMSL evolution after 10,000 years cannot be excluded in our experiments due to a further adjustment of the AIS and a continued response of the deep ocean to the surface warming and resulting thermal expansion.

## 7. Discussion

Glaciers and ice caps are the second largest contributor to present-day GMSL rise (after the thermosteric component) and will continue to be a major source of sea-level rise during the next century (WCRP Global Sea Level Budget Group, 2018). At the end of the 21[st] century, the global glacier volume is projected to decrease between 21 % (MMCP2.6) and 24 % (MMCP8.5). The contribution to GMSL from glaciers and ice caps melting generally corroborate the findings by Hock et al. (2019), who found that by the end of the 21[st] century mountain glaciers and ice caps lose between 11-25 % (RCP2.6) and 25-47 % (RCP8.5)

of their volume. For scenario MMCP8.5, our results are below the average estimates at the end of the 21[st] century because of the rather low climate sensitivity in LOVECLIM. At the end of the simulations, the lower sensitivity of our glacier model to the high forcing scenarios is irrelevant and glaciers and ice caps will lose 96 - 100 % of their volume for any of the forcing scenarios. Raper and Braithwaite (2006) and Goelzer et al. (2012) found that glaciers and ice caps disappear completely for a global warming of around 4˚C by the end of the third millennium.



GMSL rise due to thermal expansion and haline contraction has a sensitivity of 0.27 to 0.68 m per ˚C, where the stronger forcing scenarios have the higher contribution due to the additional effect of haline contraction. Levermann et al. (2013) identified a similar linear relation between thermal expansion and global mean temperatures of 0.2 to 0.63 m per °C. Hieronymus (2019) assessed several coupled climate models and found an updated sensitivity of 0.51 to 0.83 m per ˚C of surface warming. The higher sensitivity is present in climate models that show an increase in AMOC strength. However, the

AMOC strength is (temporarily) decreasing in our experiments, especially in the simulations where the GrIS is melting fast. Therefore, the higher sensitivity reported by Hieronymus (2019) might be an overestimation in future scenarios where the ice sheets will melt strongly, due to the neglect of ice sheet-ocean interactions.

The largest (scenario-based) uncertainty in future multi-millennial sea level rise comes from the polar ice sheets. In our simulations, melting of the entire GrIS takes about 10,000 years for a local annual mean temperature anomaly of 2˚C with

respect to 1970-2000 AD. In higher scenarios, the GrIS needs between 8000 years for MMCP4.5 with a mean SAT anomaly of 5.1°C  and 2000 years for MMCP-feedback with a mean SAT anomaly of 9.8 °C, to disintegrate entirely.  Robinson et al. (2012) found the temperature threshold for melting the entire Greenland ice sheet to lie between 0.8 and 3.2˚C, with a long decay time for temperatures close to this threshold. Since the temperature anomaly in scenario MMCP2.6 is close to this threshold, it takes about 10,000 years in our simulations to melt the entire GrIS. According to Aschwanden et al. (2019), the

GrIS will lose between 72 % and 100 % of its volume by the end of the next millenium, using an extreme melt forcing and neglecting the temporarily cooling effect of a reduced AMOC. Other studies found that the GrIS could disappear in less than 3000 years for a constant $CO_2$ forcing exceeding 1100 ppmv (Alley et al., 2005, Driesschaert et al., 2007, Huybrechts et al., 2011), in the same range as our simulations even though the results are not entirely comparable owing to a different time evolution of the $CO_2$ and temperature forcing.

The marine-based WAIS is considered the most vulnerable part of the AIS and collapses for cumulative $CO_2$ emission exceeding ~1100 GtC. The Wilkes subglacial basin in East Antarctica becomes ice-free in scenarios MMCP-break, MMCP8.5 and MMCP-feedback. The Aurora Basin also loses its ice cover for scenario MMCP-feedback. The ice sheet respectively contributes 12 m, 19.6 m and 27 m to sea level following scenarios MMCP-break, MMCP8.5 and MMCP-feedback (Figure 4). Our numbers for MMCP8.5 are comparable to the study by Golledge et al. (2015), equally based on an extension of the

ECP8.5 scenario with constant forcing after 2300 AD. They found that AIS retreat over the Wilkes and Aurora basins would contribute up to 11.4 m to GMSL after 5000 years and 15.7 m after 50,000 years. In the simulations of Winkelmann et al. (2015), the grounding line retreats significantly along the WAIS for a cumulative emission of 1000 GtC, while in East Antarctica the retreat is most rapid for the Wilkes and Aurora subglacial basins and initiates for a cumulative $CO_2$ emission of 2500 GtC. Considering a cumulative $CO_2$ emission of 10,000 GtC, Winkelmann et al. (2015) even found that the Antarctic ice

sheet could nearly disappear entirely. However, that is an extreme scenario at the upper end of potential carbon reserves and resources available for combustion, which we did not consider in the present paper.



Even though the resolution of the ice sheet models and the climate model is relatively coarse, especially to simulate the retreat along the outlet glaciers in Greenland and the grounding line retreat in the most sensitive Antarctic regions such as Thwaites or Pine Island glaciers in West Antarctica, the model is performing quite well to simulate the ice sheet response on the millennial
timescale. Marine terminating glaciers along the Greenland ice sheet will retreat in the coming decades and the SMB will dominate the Greenland mass loss. The Antarctic ice sheet model run at 20 km resolution is too coarse to simulate fast flowing glaciers in great detail and the omission of sub-grid scale mechanisms makes the model less sensitive on the short-term to basal melting (Levermann et al., 2020, Seroussi et al., 2020). However, the grounding line retreats on land in Antarctica for most scenarios and also here, the influence of SMB processes becomes more important. Because of the rather low contribution
of the AIS to sea-level by 2300 AD for scenario MMCP2.6 and the lower sensitivity to grounding line retreat compared to other ice sheet models (Levermann et al., 2020), we assume that our lowest sea-level change value is a conservative estimate.

It should be noted that our ice sheet models do not include hydrofracturing. Hydrofracturing may significantly speed up Antarctic ice sheet decay as achieved by the Marine Ice Cliff Instability (MICI) mechanism (Pollard et al., 2015). DeConto and Pollard (2016) find Antarctic ice sheet volume losses equivalent to a freshwater input into the surrounding ocean in excess
of 1 Sv, much larger than the peak freshwater input in our simulations of 0.15 Sv. However, the MICI is controversial for its large contribution to sea level already at the end of the 21$^{st}$ century (e.g. Edwards et al., 2019).

Clark et al. (2016) found the AIS to be the largest contributor to sea level rise after 10,000 years for all of the scenarios they considered. Moreover, it is suggested that the sensitivity of GMSL change to atmospheric $CO_2$ lowers for higher cumulative $CO_2$ emissions with a logarithmic relation between both. For the first 2 millennia, our results show a similar behaviour, but at
the end of the 10,000 years simulations, GMSL increased more for the higher emission scenarios and a stronger non-linear relationship between GMSL and cumulative $CO_2$ emissions was established. Part of the discrepancy can be explained by the difference between scenario MMCP8.5 and MMCP-feedback. Both scenarios assume the same peak $CO_2$ values, but the latter adds more carbon to the atmosphere after 10,000 years due to the methane emission feedback. Another difference with our experiments is that we include the albedo-temperature feedback, which gains importance once the Antarctic ice sheet retreats
inland.

Oppositely to previous modelling studies investigating the Antarctic ice sheet evolution on a multi-millennial timescale (Winkelmann et al., 2015, DeConto and Pollard, 2016, Clark et al., 2016), we include the two-way feedbacks between the ice sheets, the atmosphere and the ocean. Large amounts of freshwater – due to melting of the ice shelves and grounded ice from Antarctica – enter the Southern Ocean and impact on Antarctic Bottom Water (AABW) and sea ice formation (Swingedouw
et al., 2008, Goelzer et al., 2016a, Goelzer et al., 2016b). On the other hand, freshwater fluxes from melting the GrIS reduce strongly the NADW formation and weaken the AMOC. The interhemispheric see-saw effect is understood as weakening NADW formation leading to a reduced transport of cold water to the Antarctic leading to a warming of the Antarctic continent (Stocker, 1998). However, in our future warming simulations, the AIS is also melting and freshwater fluxes act as a negative





feedback by limiting the ocean temperature increase locally (Figure 11).  The observed limited warming in our simulations is

supported by a study from Swingedouw et al. (2009) who found regional cooling of up to 10 ˚C following a freshwater release

of 1 Sv in the Southern ocean. In contrast to other studies that suggest an increase in sea-ice area due to large freshwater pulses

(Swingedouw et al., 2008; Goelzer et al., 2016b) where the ocean stratification increases beneath the ocean surface leading

ultimately to ice shelf melting at depth (Golledge et al., 2014). This positive feedback is not observed in our simulations, where

the sea-ice area (and volume) declines strongly in all our experiments and almost all sea-ice disappears for the three highest

forcing scenarios at the end of the first millennium (Figure 9).

## 8. Conclusions

In this paper we have assessed multi-millenial sea-level change projections having fully interactive ice sheet components as

obtained within the Earth System Model of Intermediate Complexity LOVECLIM. Global mean sea level is shown to

continuously rise during the next 10,000 years for all scenarios considered. These build further on IPCC RCP/ECP scenarios

with emission pathways ranging from temporarily negative emissions to burning ~5000 GtC with the addition of a methane

emission feedback. The sea-level rise is thus found to be irreversible on a 10,000 year time scale, even for the lowest forcing

scenario based on extending scenario RCP2.6 with zero emissions.

The interactive nature of our model study on the multi-millennial timescale is important to simulate most climatic feedbacks

in the real world in an accurate way. Ice sheet-ocean interactions results in temporarily large perturbations in the ocean

circulation with freshwater fluxes from the GrIS leading to a strongly reduced AMOC strength and local cooling. Apart from

the local cooling effect and its influence on atmospheric temperatures in the vicinity of the Greenland ice sheet, our findings

suggest that the reduced AMOC also lowers the contribution from the steric component to GMSL. Freshwater fluxes from the

AIS lead to reduced oceanic warming in the vicinity of Antarctica and relatively low basal melt rates. Ice sheet - atmosphere

interactions on a multi-millennial timescale are important to take into account because of the SMB-elevation feedback that

sets-in when the ice sheet starts to melt and the surface elevation decreases.  The strong albedo-temperature feedback inititiates

when the ice sheets retreat on land, where temperatures increase due to an increase in the tundra-like surface type (in addition

to the effect of a recuded  sea-ice area). As a consequence, surface melting accelerates once a critical area of land becomes

ice-free.

It is found that the Greenland ice sheet will melt entirely over the next 10,000 years, but the rate of sea-level rise is determined

by the forcing scenario. Oppositely, the fate of the Antarctic ice sheet is largely dependent on the future greenhouse gas forcing

scenario considered. For the lowest forcing scenario, there is  only a limited retreat of the grounding line in West Antractica

and the East Antarctic ice sheet remains mostly unaffected, resulting in a limited sea-level contribution of 1.6 m. For the

highest forcing scenarios, the West Antarctic ice sheet will collapse entirely and there is significant marginal ice sheet retreat

along the East Antarctic ice sheet with a total volume loss of around 27 m SLE.  The combined contribution of thermal





expansion and haline contraction continues to contribute to sea level rise as long as there is a freshwater influx, even though the bulk of the oceanic heat increase takes place in the first centuries after the steep rise in temperatures. It is the only component contributing to sea level that reached a peak during the coming millennia and decreases slowly towards the end of the simulations (only for the two lowest scenarios) following a slight global cooling trend after a few thousand years. Glaciers and ice caps are the smallest component in the sea level budget and disappear relatively fast during the coming centuries.

We have identified the existence of a threshold in total cumulative $CO_2$ emissions for which GMSL is dominated by melting of the Greenland ice sheet or by GMSL contributions from the Antarctic ice sheet In our simulations, GMSL rise is dominated by melting of the Greenland ice sheet for a total cumulative $CO_2$ emission of up to 1100 GtC (after 2000 AD),. Under these scenarios, GMSL is found to quasi stabilize between 8.8 and 15 m above current levels. For the higher emission scenarios, the AIS contribution exceeds the GrIS contribution after 1000 years due to grounding line retreat and surface melting in East

Antarctica and total GMSL rise after 10,000 years ranges between 22 and 38 m. Near the end of the simulations, the rate of sea-level change decreases to values below 0.05 m per century for all forcing scenarios and sea-level approaches a semi-equilibrated state.

**Appendix A: Construction of the carbon dioxide concentration scenarios**

We assume that an impulse response function (IRF) represents the time-dependent abundance of a gas after an additional

emission pulse (Joos et al., 2013; Maier-Reimer and Hasselmann, 1987). After the peak concentration, the $CO_2$ concentration decreases exponentially reaching a short-term equilibrium $CO_2^{equi_s}$ (after 1000 years) and a long-term equilibrium $CO_2^{equi_l}$ (after 10,000 years) based on Eq. (1) and Eq. (2) (Solomon et al., 2009). The time constants used for the two exponentials do not have a process based meaning, but are fitting parameters to best represent the wide range of IRFs present in the literature. The peak airborne fraction ($AF^{peak}$), the short-term stabilization level $CO_2^{equi_s}$ and the long-term stabilisation level $CO_2^{equi_l}$ (*

can be replaced by l and s in Eq. (2) ) represent the different theoretical neutralization processes.

$$CO_2(t) = CO_2^{equi_l} + \left[ (CO_2^{peak} - CO_2^{equi_s})e^{-\lambda_s t} + (CO_2^{equi_l} - CO_2^{equi_s})e^{-\lambda_l t} \right] \quad (1)$$

Where:   $CO_2^{equi_s}$ = equilibrium $CO_2$ concentration after 1000 years

$CO_2^{equi_l}$ = equilibrium $CO_2$ concentration after 10,000 years
$CO_2^{peak}$ = peak $CO_2$ concentration
$\lambda_s$ = short term decay rate
$\lambda_l$ = long term decay rate
t = time (between peak concentration and 12,000 AD)

$$CO_2^{equi_*} = \frac{AF^{equi_*}}{AF^{peak}} \left( CO_2^{peak} - CO_2^0 \right) + CO_2^0 \quad (2)$$

455        Where:        $CO_2^0 = 280$ ppmv



$AF^{equi_*}$ = equilibrium airborne fraction
$AF^{peak}$ = peak airborne fraction

The instantaneous or peak airborne fraction ($AF^{peak}$) is the atmospheric $CO_2$ peak concentration as a percentage of the total
released $CO_2$ (Archer and Brovkin, 2008). $AF^{equi}$ measures the fraction of emitted $CO_2$ that remains in the atmosphere after
1000 years ($AF^{equi}_s$) and 10,000 years ($AF^{equi}_l$), respectively. The main principle is that the more $CO_2$ is emitted in the
atmosphere, the lower the capacity of the ocean to buffer the excess of carbon due to the limited size of the ocean (Archer et
al., 2009a). Dissolved carbon in the ocean consists of bicarbonate ($HCO_3^-$) and carbonate ions ($CO_3^{2-}$). The latter buffers the
ocean against $CO_2$ invasion, but is less abundant than bicarbonate. Depletion of the bicarbonate ions starts with increasing
atmospheric $CO_2$ concentrations. As a consequence, the buffering capacity of the ocean decreases with higher $CO_2$ atmospheric
injections (Archer and Brovkin, 2008).

The airborne fractions are chosen in accordance with the values present in the literature (Table A1) and extrapolated for the
scenarios that do not correspond to the $CO_2$ emission pulses (Table A2). The peak airborne fraction after a pulse injection of
$CO_2$ is 100 % and therefore not completely comparable with the more gradual injections as assumed by the MMCP scenarios.
Therefore, these peak airborne fractions are scaled to the peak airborne fractions for a slower injection of $CO_2$ where the peak
atmospheric $CO_2$ value is between 50 and 70 % of the real atmospheric $CO_2$ emissions (Archer and Brovkin, 2008).

**Author contribution**

PH and HG designed the experiments and JVB carried them out. HG and PH developed the model code. JVB prepared the
manuscript with contributions from all co-authors.

**Acknowledgements**

We acknowledge support through the Belgian Federal Science Policy Office within its Research Programme on Science for a
Sustainable Development under contract SD/CS/06A (iCLIPS) and the Belgian National Agency for Radioactive Waste and
enriched Fissile Material (ONDRAF/ NIRAS). Heiko Goelzer has received funding from the programme of the Netherlands
Earth System Science Centre (NESSC), financially supported by the Dutch Ministry of Education, Culture and Science (OCW)
under grant no. 024.002.001.





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



**Table 1: Sea-level contribution from the different components contributing to GMSL change for scenario MMCP2.6,**
**MMCP4.5, MMCP6.0, MMCP-break, MMCP8.5 and MMCP-feedback shown in meters and as a percentage of**
**GMSL change.**

|  | Antarctic ice sheet | Greenland ice sheet | Steric component | Glaciers and ice caps | GMSL rise |
|---|---|---|---|---|---|
| MMCP2.6 | 1.6 m (17.4 %) | 7.0 m (76.1 %) | 0.4 m (4.3 %) | 0.23 m (2.5 %) | 9.2 m |
| MMCP4.5 | 1.9 m (18.8 %) | 7.3 m (72.3 %) | 0.7 m (6.9 %) | 0.23 m (2.3 %) | 10.1 m |
| MMCP6.0 | 6.6 m (42.9 %) | 7.3 m (47.4 %) | 1.3 m (8.4 %) | 0.24 m (1.6 %) | 15.4 m |
| MMCP-break | 12.0 m (54.8 %) | 7.4 m (33.8 %) | 2.3 m (10.5 %) | 0.24 m (1.1 %) | 21.9 m |
| MMCP8.5 | 19.7 m (65.4 %) | 7.4 m (24.6 %) | 2.8 m (9.3 %) | 0.24 m (0.8 %) | 30.1 m |
| MMCP-feedback | 26.8 m (71.7 %) | 7.4 m (19.8 %) | 3.0 m (8.0 %) | 0.24 m (0.6 %) | 37.4 m |

**Table A1: Estimates of $CO_2$ airborne fractions available in the literature 1000 and 10,000 years after a pulse of $CO_2$**
**into the atmosphere.**

| Reference | $CO_2$ emission pulse (GtC) | AF (1000 year) | AF (10,000 year) |
|---|---|---|---|
| (Archer and Brovkin, 2008) | 1000 | 29 % | 14 % |
|  | 5000 | 57 % | 26 % |
| (Archer et al., 2009a) | 1000 | 24-31 % | 10-21 % |
|  | 5000 | 32-62 % | 14-32 % |
| (Eby et al., 2009) | 1000 | 30 % | 17 % |
|  | 5000 | 60 % | 30 % |
| (Joos et al., 2013) | 100 | 10-21 % | / |
|  | 100 (+350) | 20-30 % | / |
|  | 5000 | 30-60 % | / |








**Table A2: Average equilibrium airborne fractions for total emissions of around 500 to 5000 GtC according to the values given in Table A1. The equilibrium airborne fraction for 2000 GtC is extrapolated from the average equilibrium airborne fraction for the 1000 GtC and 5000 GtC scenarios. The instantaneous airborne fraction is the airborne fraction reached after one year ($AF^{peak}$). Airborne fractions after 1000 and 10,000 years are given. The columns highlighted in bold give the fractions of atmospheric $CO_2$ still present 1000 and 10,000 years after the peak concentration. The numbers in the first column denote the cumulative emissions $CO_2$ emissions with respect to pre-industrial.**

|  | $AF^{peak}$ | $AF_s^{equi}$ (1000 years) | $\dfrac{AF_s^{equi}}{AF^{peak}}$ | $AF_l^{equi}$ (10,000 years) | $\dfrac{AF_l^{equi}}{AF^{peak}}$ |
|---|---|---|---|---|---|
| 461 GtC – **MMCP2.6** (low) | 50 % | 25 % | **50 %** | 12 % | **24 %** |
| 1361 GtC – **MMCP4.5** (moderate) | 55 % | 28 % | **51 %** | 15 % | **27 %** |
| 2234 GtC – **MMCP6.0** (moderate) | 60 % | 33 % | **55 %** | 18 % | **30 %** |
| 3393 GtC – **MMCP-break** (high) | 65 % | 40 % | **62 %** | 22 % | **34 %** |
| 5288 GtC – **MMCP8.5** (high) | 70 % | 47 % | **67 %** | 25 % | **36 %** |
| 5888 GtC – **MMCP-feedback** (high) | 70 % | 55 % | **78 %** | 33 % | **47 %** |

**Table A3: Different $CO_2$ scenarios used in the simulations with their time of peak concentration, the values for the peak concentration, the mean concentration over the next 10,000 years and the equivalent total emissions (calculated as in Meinshausen et al. (2011)). The total cumulative emissions are an approximation for the RCP scenarios (the numbers give the total fossil fuel cumulative $CO_2$ emissions expressed in GtC for the period 2000-2300 AD). These numbers are on top of the cumulative $CO_2$ emissions before 2000 AD given in brackets (about 270 GtC; Ciais et al. (2013)).**

| Scenario | Time of peak concentration | Peak concentration (ppmv) | Mean concentration (ppmv) | Cumulative emissions (GtC) |
|---|---|---|---|---|
| **MMCP2.6** | 2053 | 443 | 305 | 191 (+270) |
| **MMCP4.5** | 2130 | 543 | 358 | 1091 (+270) |
| **MMCP6.0** | 2150 | 752 | 431 | 1964 (+270) |
| **MMCP-break** | 2150 | 1429 | 674 | 3723 (+270) |
| **MMCP8.5** | 2250 | 1962 | 918 | 5018 (+270) |
| **MMCP-feedback** | 2250 | 1962 | 1088 | 5618 (+270) |




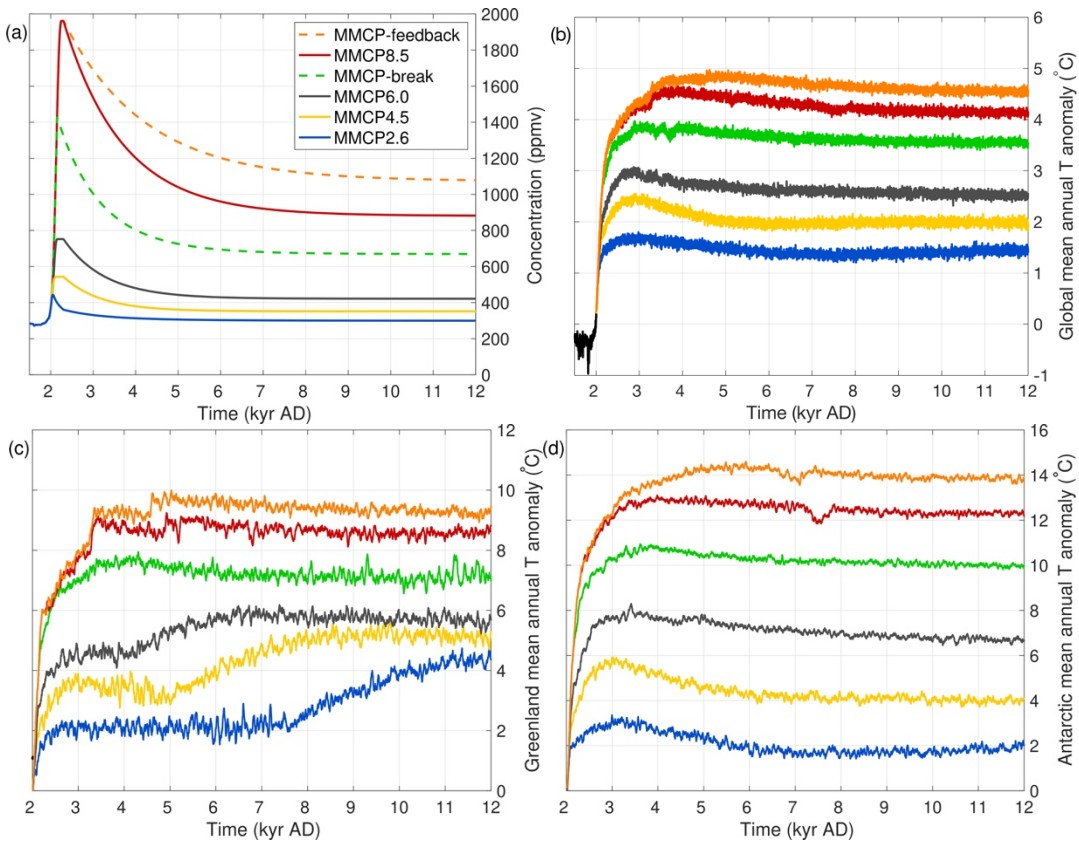

**Figure 1: (a) Atmospheric CO₂ concentration scenario from 1500 AD to 12,000 AD for the six different forcing scenarios. The four RCP-based scenarios are given in solid lines, while the two additional scenarios are shown as dashed lines. (b) Global mean temperature change for the historical period 1500-2000 AD and the future projections (c) Greenland mean temperature change and (d) Antarctic mean temperature change for the six forcing scenarios. All temperature projections are shown as an anomaly to a reference period 1970-2000.**



775



Figure 2: (a) Greenland ice sheet configuration at 3,000 AD, 7,000 AD and 12,000 AD using scenario MMCP2.6 and
(b) MMCP-feedback. Main mass balance components explaining the change in ice sheet geometry for scenarios
MMCP2.6 (c) and MMCP-feedback (d).

1["





**Figure 4: The contribution of the different components to GMSL rise during the next 10,000 years for each of the forcing scenarios. The ice sheet geometry of Greenland and Antarctica is shown at the end of the simulations.**

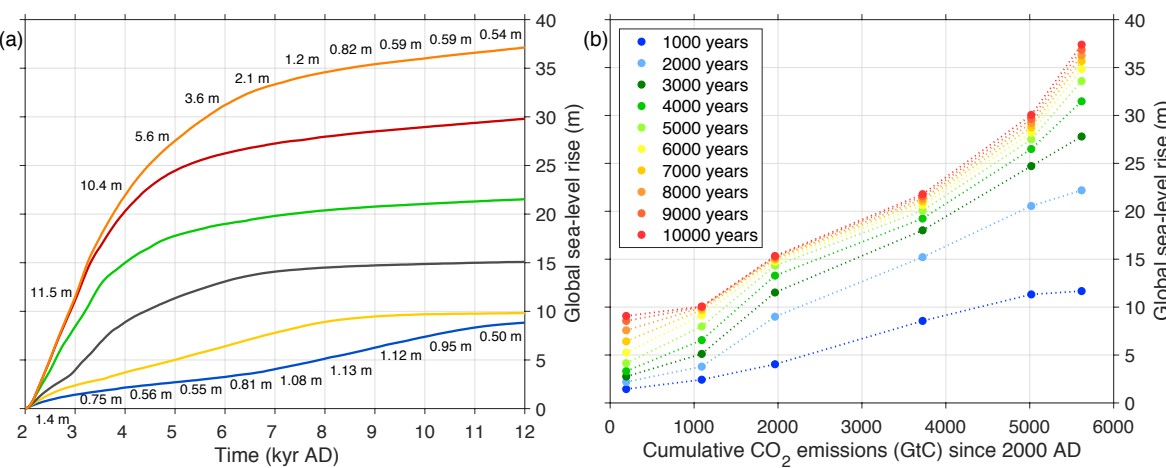

**Figure 5: (a) GMSL rise as the sum of the contributions from the AIS, the GrIS, glaciers and ice caps, and thermal expansion and haline contraction. The rate of sea level rise during each millennium is indicated for the lowest and highest scenarios. (b) GMSL rise for a given cumulative CO₂ emission at the end of each millennium until 12,000 AD.**


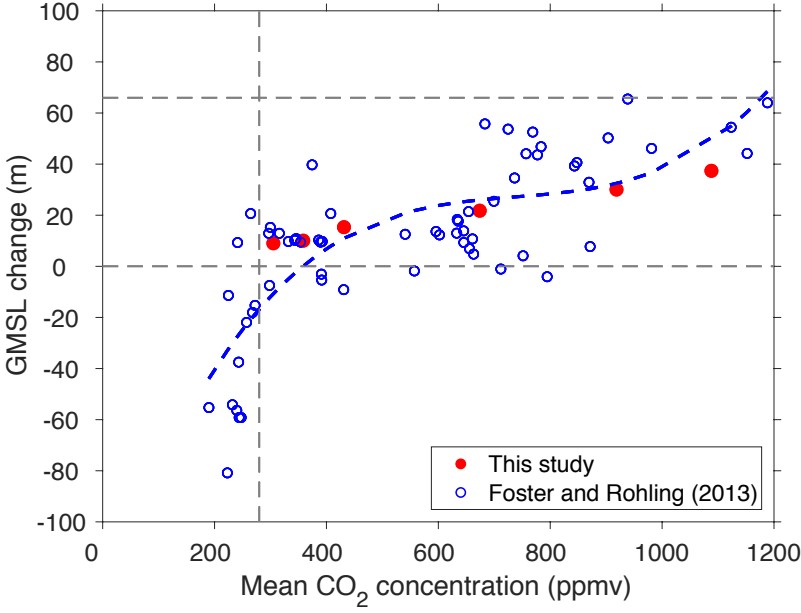

**Figure 6: Semi-equilibrated GMSL change projections at 12,000 AD compared to the geological record of sea-level high stands (Foster and Rohling, 2013) for a given atmospheric CO₂ concentration. The atmospheric CO₂ concentrations**
**used for the projections of GMSL are averaged over the 10,000-year simulation. The horizontal line at 0 m GMSL change represents the present-day situation and the horizontal line at +65 m is equivalent to melting all land ice on Earth.**





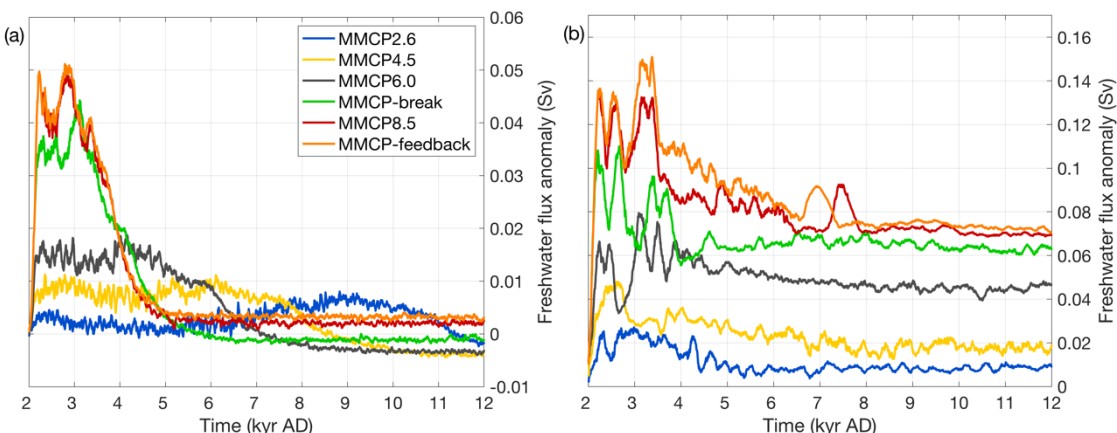

**Figure 7: Freshwater flux anomalies with respect to 1970-2000 from the Greenland ice sheet (a) and the Antarctic ice sheet (b) for the six different forcing scenarios.**

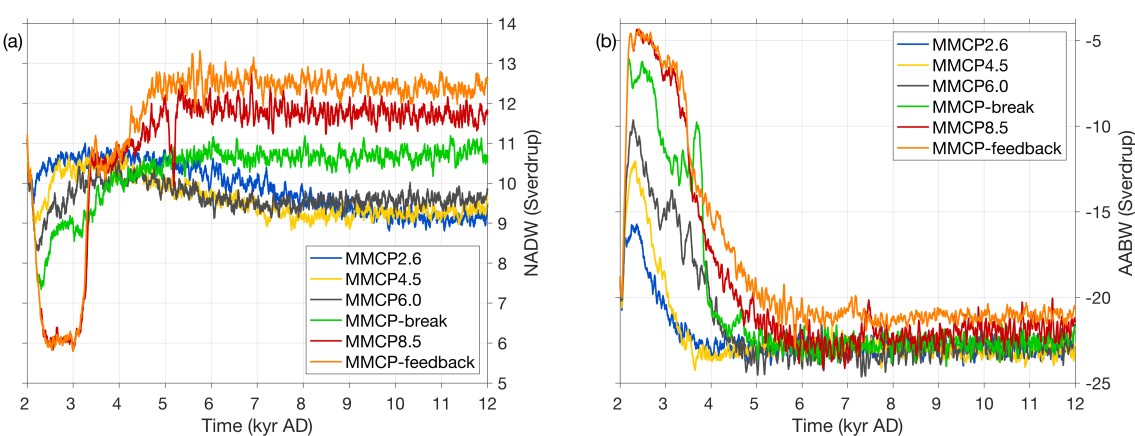

**Figure 8: Mean annual strength of the North Atlantic Deepwater (NADW) (a) and the Antarctic Bottom Water (AABW) (b) formation from 2000 AD to 12,000 AD. AABW is defined as the maximum of the global meridional overturning streamfunction in the bottom cell and NADW as the maximum of the meridional overturning circulation in the Greenland, Iceland and Norwegian Seas.**





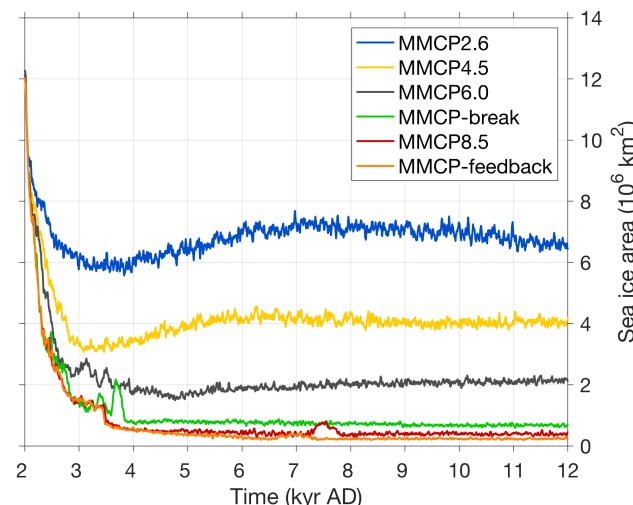

**Figure 9: Mean sea-ice area evolution in the Southern Ocean for the six different forcing scenarios.**




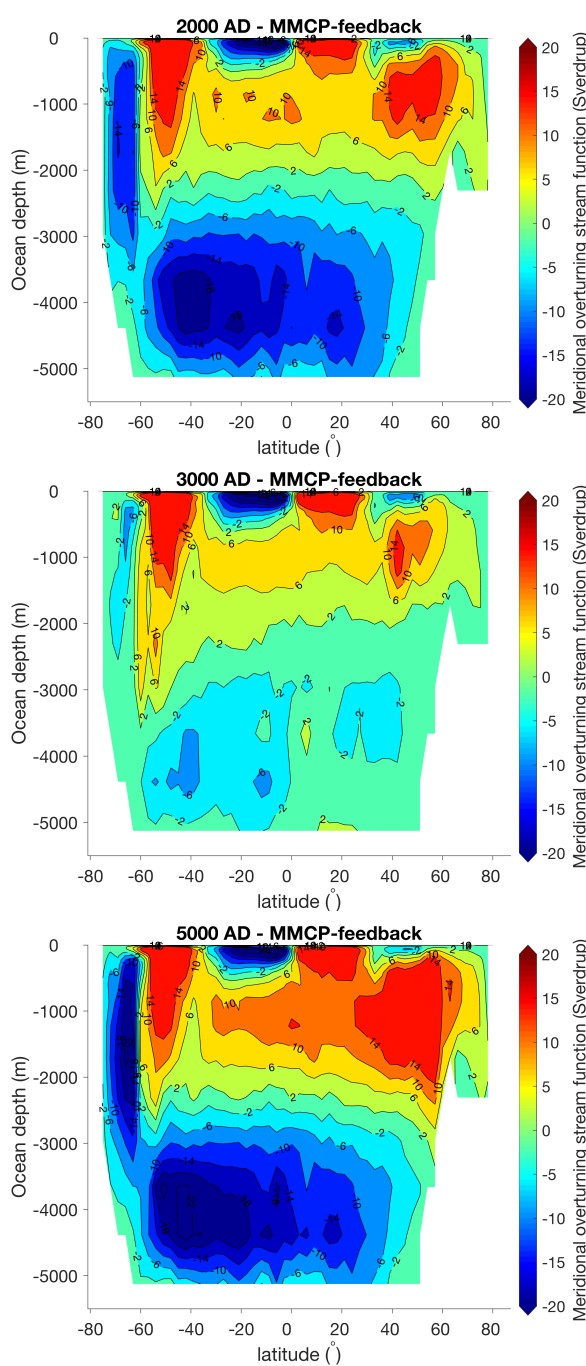


**Figure 10: Meridional overturning stream function for the global ocean at 2000 AD, 3000 AD and 5000 AD using forcing scenario MMCP-feedback.**










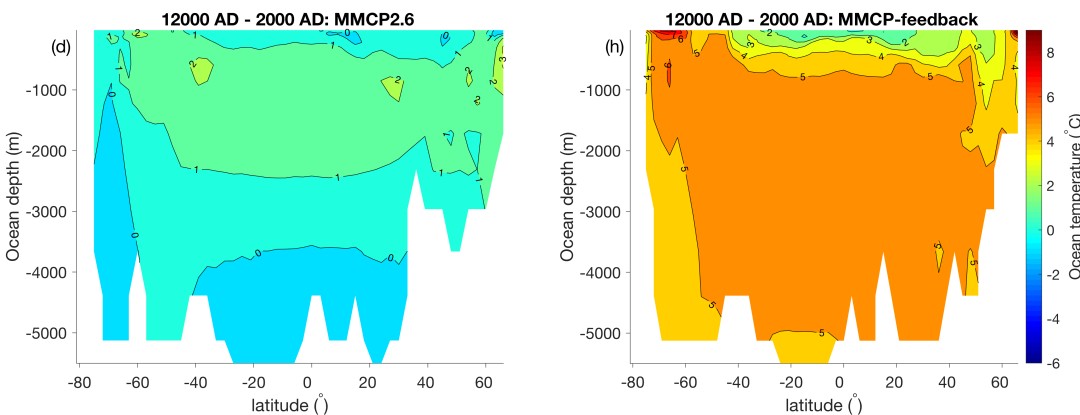

**Figure 11: Ocean temperature anomaly across the Atlantic ocean using scenario MMCP2.6 at 2100 AD (a), 2300 AD (b), 7000 AD (c), 12,000 AD (d) and using scenario MMCP-feedback at 2100 AD (e), 2300 AD (f), 7000 AD (g), 12,000 AD (h).**
