# Peer review of "Semi-equilibrated global sea-level change projections for the next 10,000 years"

_Earth System Dynamics, 2020_

## Referee Comment (RC1) · Anonymous Referee #1 · 21 May 2020

The authors used LOVECLIM to explore global sea-level change due to melting of land ice combined with steric sea effects during the next 10,000 years. They adopted the scenarios following the Extended Concentration Pathways with no carbon dioxide emissions after 2300. They found that the change in global mean sea level ranges from 9.2 m to more than 37 m after 10,000 years. The Greenland ice sheet nearly disappear for all forcing scenarios while the Antarctic ice sheet contributes about 1.6 m and up to 27 m to sea level for the lowest and higher forcing scenario, respectively.

This study investigates multi-millennial semi-equilibrated sea-level rise, which potentially contributes to our understanding of future sea level change beyond centurial timescales. Thereby, I would like to support publication after minor revisions.

First, the authors may want to show the global maps of surface air temperature, surface

winds and precipitation changes, especially over the Greenland and Antarctic ice sheet regions. Given the three-layer atmosphere model in LOVECLIM, how does LOVECLIM tackle surface air temperature and surface winds changes considering boundary layer processes? Also, how do precipitation changes affect mass balance of ice sheet and hence modulate the melting of ice sheet? How do surface wind changes drive the drifts of the Arctic and Antarctic sea ice and also drive the ocean circulation, like the Deacon Cell in the Southern Ocean?

Besides, I am confused by the AMOC change in the simulations. Why does AMOC recover after a temporary almost shutdown, and even overshoot in the higher forcing scenarios? Is AMOC mono-stable in LOVECLIM?

––––––––––––––––––––––––––––––––––

---

## Referee Comment (RC2) · Paolo Scussolini (Referee) · 4 Jun 2020

General comments

The manuscript deals with the interesting and conceptual question of the evolution of sea levels in the coming millennia, and employs a Earth system model of intermediate complexity that is adequately equipped to address the question.

The piece is fairly well-written, and the structure of the sections is suited to present and contextualize this set of results. The set of experiments with the extension of the RCP scenarios and the two additional scenarios crafted by the authors is well planned. Also, the implementation of the impulse response function seems well executed. The discussion of the results in the light of evidence and modeling of past, present and fu-

ture sea levels is comprehensive and updated to the latest literature. The explanations for the changes observed from each component of the model are convincingly argued. Still, a large number of mistakes and imprecise statements exist; see below a long list of suggested corrections. Although these points of attention are many, I still consider that revisions necessary for the manuscript to reach publishable form are minor, and no further experiment nor analysis is necessary.

Specific comments

The motivation for studying very long term climate evolution and equilibrium and sea levels is attempted, but it is not very convincing in my view. I think the authors could make a better case for the focus of this study: what gain does this knowledge represent for science or society? To play the devil's advocate, why not waiting a few years until we have a firmer grasp on sub-scale mechanisms of ice sheet loss, before attempting such long-term projections? The urgency of information on outcomes for the next millennia is not self-evident.

Also, the authors should acknowledge somewhere the somewhat speculative nature of the exercise. For example, it seems plausible that existing carbon capture technology will reach scalability (e.g., Wilberforce et al., 2019; https://doi.org/10.1016/j.scitotenv.2018.11.424), if not in the coming years or decades, plausibly in the coming centuries. These solutions would make it possible to actively reduce $CO_2$ levels, thus questioning the relevance of strong statements in the abstract, introduction and conclusions like 'long lifetime of atmospheric $CO_2$', 'will continue to rise on a multi-millennial timescale even when anthropogenic $CO_2$ emissions cease completely' and 'irreversible'. A qualifying statement about the uncertainty associated with these long-term outcomes could be added in (some) of those places.

I would reorganize sections, to account for the fact that section 6 is actually a part of the discussion. Maybe the discussion can be organized in two (or more) separate parts, one of which would deal with the contextualization of the results vis a vis the

geological record. In the methods (L 104): the 'application' of temperature and precipitation 'anomalies' seems very important and requires further explanation: what is done with these two variables precisely? Does this step include statistical downscaling and/or bias-correction? I second the inclusion of the additional scenarios MMCP-break and MMCP-feedback, but the motivation behind these choices should be made explicit. What type of situation and uncertainty do they aim to represent and address?

In various places in the paper, you bring attention to the process of 'haline contraction'. I am not an expert in this aspect, but I suggest more clarity should be made here. Since the phenomenon under analysis here is sea level rise and not fall, and since the ocean is made less saline by addition of freshwater and thus water is made to expand, 'contraction' seems a misleading term. Commonly, this component of steric sea level change is considered very much second order. I cannot find a quantification in this paper, but if it is indeed the case that salt changes are very minor compared to the other processes included, maybe negating special mention to them is warranted and improves clarity.

Technical corrections

Abstract L 27: is it not 'greenhouse forcing'? Climate should be a product of the climate model.

L 28-29: I am not sure it is clear what the methane feedback does in the model, and why it is only switched on for the highest emission scenario. If this is too complex to be explained in an abstract, maybe only mention in two words or leave out.

L 34-35: this sentence makes little sense with no further explanation: what can proxies tell (directly) about the future?

Introduction

L 47: the statement on the ice sheet response time will be read as if associated with empirical evidence, whereas it is based – if I am not mistaken - on a highly conceptual

one-dimensional model in a very dated study. Please qualify, and if applicable add empirical evidence or more recent science.

L 48-49: sea level change goes in both directions, so words like 'expansion' and 'contraction' seem in need to be complemented here.

L 51: arguably Greenland and Antarctica are also 'ice caps'? I suggest changing term for low-latitude mountain ice masses.

L 54-55: the authors will have very good insight in this process thanks also to their modeling work, but this formulation of the long-term adjustment of thermosteric sea level may be misleading: once surface temperatures are stable it would seem that the heat exchange between the ocean and the comparatively thin mass of atmosphere and land surface will only continue for a short time (probably not 'millennia') before becoming negligible. Please consider this also in the light of your results.

L 56: 'steric sea level change'?

L 63-64: what part of the ocean-atmosphere coupling did those studies take into account, and which did they neglect? Why 'full' ocean-atmosphere coupling is most important due to a process between the ice sheets and the oceans?

L 69: other EMICs exist beyond LOVECLIM that strike a balance between complexity and computational pragmatism. Please consider rephrasing to make more clear what are the specific merits of LOVECLIM here. Also, 'fully integrated coupling' is the same as 'full coupling'?

L 74: the reference for the ECP is necessary here. Also, 'to span the likely range in climate uncertainty' is not clear. Emission scenarios are meant to address emission uncertainties, rather than 'climate uncertainty'. Last, I don't think 'in a warming climate' reflects what you have implemented in these experiments.

Model description

L 80: the last millennium is frequently simulated also with GCMs, see Jungclaus et el. 2017 (10.5194/gmd-10-4005-2017). Transient simulations, such as across full deglaciations, or full glacial-interglacial cycles, are rather a specialty or EMICs.

L 91: consider adding specification of what you mean here by computational time, or it is rather meaningless: on how many cores/processors, are simulation executed in parallel fashion,. . .?

L 92: SLE abbreviation does not seem useful.

L 106: PDD? Also, there is no discussion of the specifics for the Greenland part of the ice sheet model.

L 127: maybe change to 'get ice sheets in equilibrium at 1500 AD with climate (forcing?)'?

L 128: 'without seeing the changes of the climate model' is not clear. Further, you refer to the ice sheet model as something external to the 'climate model', which in turn you have not defined. This is confusing as previously you implied that also the ice sheet component/model is part of the EMIC.

L 130: It is not clear why a different run is necessary to assess the drift with this 'quasi-equilibrium' run. Note that 'quasi-equilibrium' is not defined.

L 138: what do you mean by 'initial' here: are the differences in initial conditions applied at the end of the semi-equilibrium spin up, and to which component? Is this explained in the next sentence? If so, please check the terminology, i.e. are 'ensemble of five members' and 'five iterations of the reference state' the same thing reworded?

Scenario description

L 145: Are Multi-millennial concentration pathways introduced here for the first time? Please clarify in the manuscript.

L 158: Please explain that MMCP-feedback is based on RCP 8.5.

L 162: Maybe the methane release should be specified (also) as a rate here. Also, instead of 'by adding constantly CO2 after 2250 AD', please explain that in the simulation it is assumed that all released CH4 instantly converts to CO2. Please also consider whether it is necessary to argue that this instant conversion is a warranted simplification, since the reader will know that a molecule of methane exerts much more greenhouse effect than a molecule of carbon dioxide, and this process may not be negligible even if it takes place on a time frame much shorter than the overall simulation length.

L 167: referring to the figure here would seem appropriate.

L 175 on: please revise this whole paragraph, as I am not sure that I can follow properly your explanation here.

L 183: 'included in the climate forcing' here is confusing. It implies that the model at the centre of attention here is only the ice sheet part, whereas you are running coupled climate experiments, for which the orbital forcing is external forcing.

L 186: this is misleading. Solar forcing for the future is not 'unknown': its orbital part is very well known, whereas what is unknown is the evolution of solar cycles.

L 188 on: please reword to 'The following sections show...'. Also, the ensuing list is not clear, it reads as if the climate responds to the sea level change. Please reword.

L 190 and other instances: you use the term 'haline contraction', but is it the case that the ocean becomes more saline and contracts in your simulations? If not, then the term is misleading.

L 219: change to 'i.e., the difference between accumulation...'

L 220: 'for all the forging scenarios'

L 222: instead of the vague 'in a high warming scenario', please refer specifically to the scenario has you have named it.

[Figure]

In sections 4.3 and 4.4, and figure 2 and 3, there is confusion. Is SMB the same as mass balance in the figure? In the text SMB is the difference between accumulation and ablation, but the figure reports accumulation, calving and runoff, and apparently not the SMB, nor the amount of ice at any given moment, which would also seem a useful metric. As a suggestion, the tiles of sections 4.3 and 4.4 could mention also ocean currents, since these results are also prominent there.

L 238: punctuation is missing.

L 259: since glacial isostatic processes are included, consider mentioning this when mentioning the model description. Are these processes carried out by the is the land-surface module, which if I am not mistaken is part or ECBilt?

L 282: although an asymptotic behavior seems to emerge for all scenarios, it would be interesting to mention (and later discuss?) the late convergence between SLR of the 2.6 and 4.5 scenarios, which seems unexpected to me.

L 291: after 10000 years of simulation, or after year 10000 of the simulation?

L 298: what do you mean by 'inferred'? is it reconstructed/measured by use of/in proxies? In the next sentence, I suggest adding mention of which two periods those combinations of sea level and $CO_2$ concentrations refer to. Next sentence still: I suggest mentioning here Figure 6. On figure 6: it is bizarre that it does not show the -120 m sea level for 180 ppm mentioned in the text, I guess because the figure only uses Foster and Rohling 2013 and that work did not include such low stand. Nevertheless, because that extreme is irrelevant to the range of values here, it seems acceptable. Further in fig. 6: hat is the vertical line, pre-industrial concentration? While mentioned in the text, there is no red line for the linear fit in figure 6.

L 315: eliminate 'both of'. In this context, a test to assess the likelihood that the data from this study belong to the distribution of data from the geological records would seem informative.

L 318: thermal expansion seems quite flat for all scenarios, 10000 years into the simulations, so this can't plausibly be a contribution to further sea level rise that's meaningful for the scale of fig. 6.

Discussion

L 321: this does not seem a suitable reference for future sea level rise. Many good references for this have already been cited in the introduction. Further, it's puzzling to see the discussion open with a contributor to SLR that is not the most relevant at present and by far not in the time-scales of this study.

L 322: are RCPs more appropriate than MMCPs here?

L 330: are these numbers on the steric contribution from this study? Please clarify.

L 333: what do you mean by 'updated', is this a different generation of climate models?

L 335: verb tense is wrong.

L 336-337: it should be stated more clearly that the models included in the reference cited do not have coupling between ocean and ice-sheets (if that is the case).

L 339: 'local annual mean temperature' and 'mean SAT' seem to mean the same thing here, but different terminology is confusing.

L 389: it is not clear whether the impacts on AABW and sea ice formation are from this study or from the references listed.

Conclusion

L 406: Related to one of the main points above, it seems inappropriate to state that SLR is irreversible. That is, from your scenarios and results it appears irreversible absent active anthropogenic carbon sequestration, i.e., under the debatable assumption of no anthropogenic alteration of the carbon cycle beyond the atmospheric emission of $CO_2$ and methane.

L 408: to 'simulate' 'in the real world' seems an oxymoron.

L 431: change to 'or the Antarctic ice sheet.'

Fig. 1, unlike other figures, has the two additional scenarios in dashed lines instead of solid lines. Whereas the reason is given in the caption, this lack of consistency across figures and panels is not advisable.

Fig. 5a lacks the legend for the scenarios, which is all the more confusing because colors in 5b are used to another purpose. Also, the caption may be confusing: is this GMSL due to all relevant processes, or is it necessary to list them all here making the reader think that maybe some other process is left out? Finally, I am not sure that panel 5b is the most efficient way to show the timing difference between (cumulative) emissions and sea level change. A plot of those two quantities against time would have several benefits compared to 5b: it would show the timing aspect more clearly, it would show the scenarios, and would be much easier to read. Please take this as a suggestion.

[Figure]

---

## Referee Comment (RC3) · Anonymous Referee #3 · 23 Jun 2020

**Review of the paper entitled: "Semi-equilibrated global sea-level change projections for the next 10,000 years", by Jonas Van Breedam, Heiko Goelzer, Philippe Huybrechts.**

In this paper, the authors aim to investigate the evolution of Greenland, Antarctica ice sheets and minor components of sea-level rise over a long period (10ky) using an integrative strategy.
A model of intermediate complexity, LOVECLIMv1.3, is coupled with ice sheets models (Greenland and Antarctica) that enable them to test different $pCO_2$ and methane scenarios for the period of 1000 to 10000 years.
They investigate indeed the response of the Earth climate system to a large but short lasting perturbation. It is necessary to run long simulations that account for long time response of deep-ocean, ice sheets and $CO_2$ evolution.
The authors first describe the tool they used, the originality of which is to account for feedback between atmosphere/ocean and ice sheet, then they describe the scenarios they chose and they finally present their results in terms of different contributions.
The paper is well written and the issues are interesting, nevertheless there is room for improvements on several points.

**1. Discussion of the limitation of the study and its possible consequences:**
The scenarios are prescribed from an initial perturbation based on the four RCPs of IPCC scenarios and 2 supplementary scenarios. This paper represents an improvement compared to previous studies because feedbacks between climate and cryosphere are accounted for. The discussion on the $CO_2$ evolution, which is driven by different anthropic pathways, is only discussed from a "mathematical" point of view. The authors should discuss the limitation of such an approach. Indeed there are also interactions between carbon cycle and ocean and interactions between vegetation and carbon cycle, which are not limited to permafrost and clathrate destabilizations, and which are not accounted for. For instance, a dynamical vegetation model could be useful to account for the effect of desertification (albedo and water cycle). Another issue that is not discussed is the long-term climate evolution. Indeed all scenarios depict a complete melting of the GRIS. The simulated climates are similar to Pliocene climate and thus the impact of orbital forcing/orbital parameters may be drastically modified in comparison with Quaternary large glacial/interglacial oscillations.

**2. Computation of grounding line evolution with coarse-grid modeling:**
The response of the grounding line is very important and should be discussed in more details because it is difficult to compute it using climate change simulattion by a coarse-grid model. For instance, to test the capability of their model in order to compute correctly changes in grounding line, the authors could use the last deglaciation and they should validate their model over such a period.

**3. Parametrization and scenarios:**
Line 114-115: in addition to the fact that the authors wrote P11 in the text and P71 in the Table S3, this part of the paper is very unclear to me. The authors chose one set of parameters, and with this set, they provide 6 different simulations. All these scenarios do not need to be run again. But for the 2 extreme ones (RCP2.6 and feedback), we would like to have the result when using different parametrizations as far as LOVECLIMv1.3 needs parameterization to compensate the approximation made.
Indeed we would like to know how much the results are dependent on the parametrization.
Moreover, the scenario using methane from clathrate emission is important not only in terms of quantity of greenhouse gases emitted but these emissions could last several kyears, which is the duration of Paleocene-Eocene thermal maximum. The authors should explain the reason of their choice.
If the authors account for these main comments and more analytical comments below, I consider that this paper is a valuable contribution to an important issue and should be published in Earth System Dynamics.

**More detailed comments:**
- Title: the author use the term "semi-equilibrated", which is never clearly defined. I would prefer "quasi-equilibrium" but anyway the author should give an objective criteria for this term.
- Abstract: what is a semi-equilibrium? Over 10ka it may be important to account for astronomical forcing, especially for precession cycle (the period of interest here 1-10 ky is half the duration of precession cycle). I don't really understand the last sentence of the abstract: there is no geologic analogue for the next 10ky in Earth history - as far as I know.
How is it possible to reach more than 5800 GtC?

**1. Introduction**
The introduction if fine but 2 topics should be introduced or developed:

1. The last deglaciation, which lasts around 10 000 y, is an interesting period to validate the model used here. The authors should discuss this point, which is completely absent in the introduction.
2. The methane hypothesis should be clearly explained. There is a first short term feedback linked to the permafrost melting and a long term effect on clathrate destabilization. Concerning this second point, there are several unknowns: the quantity of methane, which is discussed by the authors but also the onset and the duration of these emissions, which they should discuss more.

**2. Model description and initialization**
The authors should clarify how they downscale the large grid of atmospheric and ocean models to high resolution ice sheet models (GRIS or AIS).
Line 105: what are the range of corrected biases for present day climate?
Line 106: is the PDD really appropriate for this study? Why did the authors use a method based on present day (cold context) rather than a method based on energy balance? Concerning the choice of parameters (Table S3) the authors should justify this choice and its possible consequences.
Line 117: the tuning of parameters on the last 500y is not really appropriate to explore large changes of cryosphere. The last deglaciation is certainly a better but more complex target for the goal of this paper.
Moreover, the authors selected a parameterization "because of its mid-range contribution to sea-level at 2100 AD and 2300 AD in comparison with recent studies". Is it a correct criteria in science to be "mid-range"? I think the authors should favor more physically based parametrizations. We also would like them to use other parameterizations for the scenarios including feedbacks.

**3. Scenario description**
Line 150-153: scenario RCP8.5 leads to a maximum PCO2 of around 5000 ppm about 20 PAL. But is there enough fossil fuel to be burnt to achieve such a value?

**4. Climate response and global sea-level budget of individual terms**
The last deglaciation is strongly nonlinear, with acceleration, as during meltwater pulse and reduced SLR during colder episodes.
The ability of the model used here to reproduce the deglaciation should be discussed.
A plot showing both terms accumulation and ablation in the different scenarios could be interesting to be depict and discuss.
For GRIS what is the ocean dynamics in North Atlantic? It seems that the AMOC recovers and is even stronger than for PD: does that mean that the GRIS could not be covered by perennial ice sheet for long periods?

**6. Long-term sea level rise in the light of the geological record**
Line 312 and Fig 6

This is an interesting comparison. Nevertheless, there should be an hysteresis between the values of melting GRIS and AIS in the simulations described here and the pCO2 values corresponding to the onset of the same ice sheets during Cenozoic due to the changes of surface albedos.

**7. Discussion**

Line 360: There are two sources of uncertainties that the authors should comment with more details:
1. The AMOC evolution because in PLIOMIP2, most of the models depicted a lower AMOC, which provided a context not favorable to the onset of GRIS
2. The authors should compare their highest scenarios to the resources available in terms of fossil fuel.
The authors pointed out the possible mechanisms that speed up the AIS melting as Marine Ice Cliff Instability MICI.
Moreover, the record of the previous deglaciation show much variability in SLR rise with acceleration and regression of the ice sheets. These simulations did not reproduce this variability in future scenarios maybe partly due to coarse resolutions. There are anyway hot debates on the prevision of SLR only concerning the end of this century and very few models are able to reproduce using transient experiments the SLR during the lest glacial/interglacial cycle. For all these reasons, the authors should be careful of uncertainties and limitations of their strategy even if it is a consistent approach for investigating long time scale.
Moreover, the evolution of the grounding line when using coarse grid model is difficult to capture and therefore, it is another limitation of the paper that the authors should comment.

**8. Conclusion**
Their main results are summarized:
A GRIS melting for all scenarios, this melting being irreversible in the time window of interest here whereas the AIS contributes to the SLR differently, depending on scenarios.
These results are consistent with a GrIS becoming perennial only when CO2 is around 300-400 ppm whereas AIS needs a higher CO2 level to become perennial around 800 ppm.
The authors should also discuss the consistency of their results when compared to the evolution of these ice sheets.

**Comments on Figures**
General comment for all the Fig. :
In fact, while GMSL rises to high values (env. 30 m) many regions that are considered in the simulation as land points shift to ocean points. By the way, the ocean/continent distribution is changing with SLR. This is certainly a minor effect for the coarse grid model for several meters. But it may be important when SLR reaches 10 to 30 meters. This effect is not accounted for in the figure and I believe also that this distribution is kept fixed in all simulations. Thus, the authors should discuss this approximation.
Fig 2: it could be interesting to show also the GRIS configuration for present day.
Fig 3: interesting to have AIS for present day. Moreover, it could be interesting to add a snapshot at 3ky with WAIS and EAIS separately.
Figure 7: the logic of the caption (a) (b) succession is not easy to follow at first sight.

---

## Author Comment (AC1) · 21 Jul 2020

**Response to Anonymous Referee #1**

The authors used LOVECLIM to explore global sea-level change due to melting of land ice combined with steric sea effects during the next 10,000 years. They adopted the scenarios following the Extended Concentration Pathways with no carbon dioxide emissions after 2300. They found that the change in global mean sea level ranges from 9.2 m to more than 37 m after 10,000 years. The Greenland ice sheet nearly disappears for all forcing scenarios while the Antarctic ice sheet contributes about 1.6 m and up to 27 m to sea level for the lowest and higher forcing scenario, respectively.This study investigates multi-millennial semi-equilibrated sea-level rise, which potentially contributes to our understanding of future sea level change beyond centurial timescales. Thereby, I would like to support publication after minor revisions.

**Author's response:** Thank you very much for the positive evaluation.

First, the authors may want to show the global maps of surface air temperature, surface winds and precipitation changes, especially over the Greenland and Antarctic ice sheet regions. Given the three-layer atmosphere model in LOVECLIM, how does LOVECLIM tackle surface air temperature and surface winds changes considering boundary layer processes? Also, how do precipitation changes affect mass balance of ice sheet and hence modulate the melting of ice sheet? How do surface wind changes drive the drifts of the Arctic and Antarctic sea ice and also drive the ocean circulation, like the Deacon Cell in the Southern Ocean?

**Author's response:** Given the numerous plots that we provided, we chose to add fields of surface air temperature change above the Greenland and Antarctic region in the supplementary information, since they are key in the long-term surface mass balance dominated melting of the ice sheets. We also added information on how the surface temperature and surface winds are computed in LOVECLIM in the model description section (section 2). Additionally, the positive degree day model (melt model) is explained in more detail and we explain how a change in precipitation affects the mass balance. We consider that the detailed discussion of the effect of surface winds on sea-ice drifts and Southern Ocean circulation is outside the scope of this study.

Besides, I am confused by the AMOC change in the simulations. Why does AMOC recover after a temporary almost shutdown, and even overshoot in the higher forcing scenarios? Is AMOC mono-stable in LOVECLIM?

**Author's response:** This is an interesting topic raised by the reviewer that we are now discussing in more detail (section 8). There is a large uncertainty whether the AMOC is mono-stable or bi-stable in climate models (GCM and EMIC's) and we gave more attention to the uncertainty that exists among climate models to simulate AMOC changes in response to freshwater fluxes.

*"The AMOC in LOVECLIM exhibits a mono-stable behaviour with a recovery to the initial state as soon as the meltwater pulse halts. An EMIC intercomparison study*

*found that 7 models show a bi-stable regime and 4 others a monostable regime following a freshwater perturbation in the North Atlantic (Rahmstorf et al., 2005). A mono-stable regime of the AMOC is simulated in most GCM's where a freshwater pulse leads to a temporary reduction of the AMOC strength and a recovery when the freshwater pulse terminates (Liu et al., 2014). However, it is speculated that a mono-stable regime might be caused by a negative salinity bias in GCM's (Mecking et al., 2017) and that a bi-stable regime would explain the rapid climatic changes during the deglaciation better (Ganopolski and Rahmstorf, 2001). The equilibrium response of the ocean suggests that the AMOC strength will also increase in response to atmospheric warming, possibly due to a decrease in sea-ice area (Jansen et al., 2018). Several studies found that the AMOC was also stronger during the mid-Pliocene, in absence of freshwater feedbacks due to ice sheet melting (Chandan and Peltier, 2017; Chan and Abe-Ouchi, 2020). Our study supports the increase of the AMOC strength when the freshwater forcing halts."*

---

## Author Comment (AC2) · 21 Jul 2020

**Response to Referee Paolo Scussolini**

General comments

The manuscript deals with the interesting and conceptual question of the evolution of sea levels in the coming millennia, and employs a Earth system model of intermediate complexity that is adequately equipped to address the question.The piece is fairly well-written, and the structure of the sections is suited to present and contextualize this set of results. The set of experiments with the extension of the RCP scenarios and the two additional scenarios crafted by the authors is well planned. Also, the implementation of the impulse response function seems well executed. The discussion of the results in the light of evidence and modeling of past, present and future sea levels is comprehensive and updated to the latest literature. The explanations for the changes observed from each component of the model are convincingly argued. Still, a large number of mistakes and imprecise statements exist; see below a long list of suggested corrections. Although these points of attention are many, I still consider that revisions necessary for the manuscript to reach publishable form are minor, and no further experiment nor analysis is necessary.

**Author's response:** Thank you very much for the detailed comments and the positive review.

Specific comments

The motivation for studying very long term climate evolution and equilibrium and sea levels is attempted, but it is not very convincing in my view. I think the authors could make a better case for the focus of this study: what gain does this knowledge represent for science or society? To play the devil's advocate, why not waiting a few years until we have a firmer grasp on sub-scale mechanisms of ice sheet loss, before attempting such long-term projections? The urgency of information on outcomes for the next millennia is not self-evident.

**Author's response:** The argument to wait a few years to get better insight in the sub-scale mechanisms of ice sheet loss is not entirely convincing. Models are continuously being improved but will never be a perfect match of reality. The reviewer possibly alludes to processes such as grounding-line mechanics or basal sliding. However on a millennial timescale, the mass loss of the Greenland and Antarctic ice sheets is mainly dominated by surface mass balance processes and feedbacks that play between the different components of the Earth's system, and less so by the details of the ice dynamics. This study also tries to investigate the effect and magnitude of these feedbacks. From a political point of view, there is a need to investigate the consequences on the very long term to inform decisions to be taken now. In our opinion, this is a valid motivation.

Also, the authors should acknowledge somewhere the somewhat speculative nature of the exercise.For example, it seems plausible that existing carbon capture technology will reach scalability (e.g., Wilberforce et al., 2019;https://doi.org/10.1016/j.scitotenv.2018.11.424), if not in the coming years or

decades,plausibly in the coming centuries. These solutions would make it possible to actively reduce CO2 levels, thus questioning the relevance of strong statements in the abstract, introduction and conclusions like 'long lifetime of atmospheric CO2', 'will continue torise on a multi-millennial timescale even when anthropogenic CO2 emissions cease completely' and 'irreversible'. A qualifying statement about the uncertainty associated with these long-term outcomes could be added in (some) of those places.

**Author's response:** We thank the reviewer for this suggestion. We took more care about the statements on irreversibility. Clearly the reviewer is right that irreversibility is a strong statement and that it depends much on future technological developments. We made clear that for our RCP-based scenarios, sea-level rise is irreversible in our simulations, but that we ignore a scenario where carbon dioxide is very efficiently extracted from the atmosphere to achieve pre-industrial CO2 levels. On the other hand, we do not exclude the use of carbon capture and storage technologies, since our lowest forcing scenario MMCP2.6 includes negative emissions during the 21$^{st}$ century.

I would reorganize sections, to account for the fact that section 6 is actually a part of the discussion. Maybe the discussion can be organized in two (or more) separate parts, one of which would deal with the contextualization of the results vis a vis the geological record.

**Author's response:** The reviewer is right that section 6 could be part of the discussion. However, we believe that section 6 is a logical bridge between the results section and the discussion of our results with respect to other studies on future ice sheet melting and sea-level rise and prefer to keep the structure as it is.

In the methods (L 104): the 'application' of temperature and precipitation 'anomalies' seems very important and requires further explanation: what is done with these two variables precisely? Does this step include statistical downscalingand/or bias-correction?

**Author's response:** The climate model output is bias corrected by applying the temperature and precipitation anomalies with respect to a reference period (1970-2000 AD). No statistical downscaling is explicitly included.

I second the inclusion of the additional scenarios MMCP-break and MMCP-feedback, but the motivation behind these choices should be made explicit.What type of situation and uncertainty do they aim to represent and address?

**Author's response:** We considered scenario MMCP-feedback to give a measure of the uncertainty that arises when global surface temperatures will increase significantly. In such a warm world, it is possible that other feedbacks arise that we wanted to account for. The reason to include MMCP-break is to have an intermediate scenario between MMCP6.0 and MMCP8.5. There is a large spread in global emissions and radiative forcing for these two scenarios in comparison to the other RCP scenarios. This information is added in the manuscript.

In various places in the paper, you bring attention to the process of 'haline contraction'. I am not an expert in this aspect, but I suggest more clarity should be made here. Since the phenomenon under analysis here is sea level rise and not fall, and since the ocean is made less saline by addition of freshwater and thus water is made to expand, 'contraction' seems a misleading term. Commonly, this component of steric sea level change is considered very much second order. I cannot find a quantification in this paper, but if it is indeed the case that salt changes are very minor compared to the other processes included, maybe negating special mention to them is warranted and improves clarity.

**Author's response:** This is a very good suggestion raised by the reviewer. After the model description, where the difference between thermal expansion and haline contraction is explained, we further refer to this combined term as the steric sea-level change component, which is the general term for thermosteric sea level change (thermal expansion) and halosteric sea level change (haline contraction).

Technical corrections

Abstract

L 27: is it not 'greenhouse forcing'? Climate should be a product of the climate model.

Indeed, this is adapted.

L 28-29: I am not sure it is clear what the methane feedback does in the model, and why it is only switched on for the highest emission scenario. If this is too complex to beexplained in an abstract, maybe only mention in two words or leave out.

The methane feedback is now explained in a few words in the abstract.

L 34-35: this sentence makes little sense with no further explanation: what can proxies tell (directly) about the future?

This sentence is removed from the abstract.

Introduction

L 47: the statement on the ice sheet response time will be read as if associated with empirical evidence, whereas it is based – if I am not mistaken - on a highly conceptual paper one-dimensional model in a very dated study. Please qualify, and if applicable add empirical evidence or more recent science.

The sentence is rephrased and a more recent reference is added.

L 48-49: sea level change goes in both directions, so words like 'expansion' and 'contraction' seem in need to be complemented here.

We believe that these concepts are explained in the next few lines.

L 51: arguably Greenland and Antarctica are also 'ice caps'? I suggest changing term for low-latitude mountain ice masses.

The large ice masses on Greenland and Antarctica are referred to as ice sheets. There is a clear distinction between ice sheets and ice caps in terms of ice extent and ice thickness.

L 54-55: the authors will have very good insight in this process thanks also to their modeling work, but this formulation of the long-term adjustment of thermosteric sea level may be misleading: once surface temperatures are stable it would seem that the heat exchange between the ocean and the comparatively thin mass of atmosphere and land surface will only continue for a short time (probably not 'millennia') before becoming negligible. Please consider this also in the light of your results.

This is a good remark from the reviewer and we changed this towards 'multiple centuries', in line with our results.

L 56: 'steric sea level change'?

Done.

L 63-64: what part of the ocean-atmosphere coupling did those studies take into account, and which did they neglect?

We thank the reviewer for this good suggestion and we made clear how the ice sheets were interacting with the climate system for these different studies.

Why 'full' ocean-atmosphere coupling is most important due to a process between the ice sheets and the oceans?

We clarified this by changing the sentence towards 'the full coupling between the ice sheets, the atmosphere and the ocean". The freshwater fluxes also have a cooling influence on the atmosphere.

L 69: other EMICs exist beyond LOVECLIM that strike a balance between complexity and computational pragmatism. Please consider rephrasing to make more clear what are the specific merits of LOVECLIM here.

The merit of LOVECLIM is the inclusion of a fully coupled ice sheet component. It is further explained in the model description that this is a rare component for EMICs.

Also, 'fully integrated coupling' is the same as 'full coupling'?

Indeed, this is changed to 'full coupling' to avoid confusion.

L 74: the reference for the ECP is necessary here. Also, 'to span the likely range in climate uncertainty' is not clear. Emission scenarios are meant to address emission uncertainties, rather than 'climate uncertainty'. Last, I don't think 'in a warming climate' reflects what you have implemented in these experiments.

The reference is added and the term 'climate uncertainty' is changed to 'emission uncertainties'.

Model description

L 80: the last millennium is frequently simulated also with GCMs, see Jungclaus et al. 2017 (10.5194/gmd-10-4005-2017). Transient simulations, such as across full deglaciations, or full glacial-interglacial cycles, are rather a specialty for EMICs.

We added the information that Earth System Models with a higher spatial resolution are expected to be able to simulate the last millennium (Jungclaus et al., 2017), in addition to the EMIC's that used to be the only tool.

L 91: consider adding specification of what you mean here by computational time, or it is rather meaningless: on how many cores/processors, are simulation executed in parallel fashion,...?

We do not have a good record of the spent core hours and therefore decided not to include specific numbers on the computational time.

L 92: SLE abbreviation does not seem useful.

We would like to use the abbreviation since it is used on line 94 and line 433.

L 106: PDD? Also, there is no discussion of the specifics for the Greenland part of the ice sheet model.

PDD is replaced by Positive Degree Day and the model is explained in more detail. The Greenland ice sheet model code is similar to previous model versions and we added references for the interested reader.

L 127: maybe change to 'get ice sheets in equilibrium at 1500 AD with climate (forcing?)'?

It is the other way around: 'to get the climate in equilibrium with the ice sheets at 1500 AD'.

L 128: 'without seeing the changes of the climate model' is not clear. Further, you refer to the ice sheet model as something external to the 'climate model', which in turn you have not defined. This is confusing as previously you implied that also the ice sheet component/model is part of the EMIC.

This sentence is rephrased towards: 'the ice sheet models in this experiment evolve very slowly for a fixed climate'. In these precursor experiments, there is not a full coupling between all climatic components, but all model components will be in equilibrium at 1500 AD after which the fully coupled model run starts.

L 130: It is not clear why a different run is necessary to assess the drift with this 'quasi-equilibrium' run. Note that 'quasi-equilibrium' is not defined.

*Our goal was to have a climate in equilibrium at 1500 AD. From 1500 AD, the model was forced in fully coupled mode with PMIP3 data. We performed the quasi-equilibrium run (the run where the climatic variables converge to a nearly constant value) to assess whether this statement was indeed true.*

L 138: what do you mean by 'initial' here: are the differences in initial conditions applied at the end of the semi-equilibrium spin up, and to which component? Is this explained in the next sentence? If so, please check the terminology, i.e. are 'ensemble of five members' and 'five iterations of the reference state' the same thing reworded?

*The initial conditions are slightly different for the boundary conditions of the atmospheric component. These initial conditions are applied at the end of the quasi-equilibrium run. The 'ensemble of five members' and five iterations of the reference state' are not the same and these sentences are rephrased for clarity.*

Scenario description

L 145: Are Multi-millennial concentration pathways introduced here for the first time? Please clarify in the manuscript.

*Yes, we explicitly mentioned now that they are introduced here.*

L 158: Please explain that MMCP-feedback is based on RCP 8.5.

*Done.*

L 162: Maybe the methane release should be specified (also) as a rate here.

*Done.*

Also,instead of 'by adding constantly $CO_2$ after 2250 AD', please explain that in the simulation it is assumed that all released $CH_4$ instantly converts to $CO_2$. Please also consider whether it is necessary to argue that this instant conversion is a warranted simplification, since the reader will know that a molecule of methane exerts much more greenhouse effect than a molecule of carbon dioxide, and this process may not be negligible even if it takes place on a time frame much shorter than the overall simulation length.

*The explanation that all $CH_4$ instantly converts to $CO_2$ was present on L166. We added the remark that this simplification neglects the short-term warming effect of methane.*

L 167: referring to the figure here would seem appropriate.

*Done.*

L 175 on: please revise this whole paragraph, as I am not sure that I can follow properly your explanation here.

Some parts of this paragraph are excluded to improve clarity and to avoid confusion. The main message is that methane and nitrous oxide have high natural emissions in a warmer world and therefore they do not seem likely to be reduced during the next 10,000 years.

L 183: 'included in the climate forcing' here is confusing. It implies that the model at the centre of attention here is only the ice sheet part, whereas you are running coupled climate experiments, for which the orbital forcing is external forcing.

This is rephrased.

L 186: this is misleading. Solar forcing for the future is not 'unknown': its orbital part is very well known, whereas what is unknown is the evolution of solar cycles.

This is rephrased.

L 188 on: please reword to 'The following sections show...'. Also, the ensuing list is not clear, it reads as if the climate responds to the sea level change. Please reword.

Done.

L 190 and other instances: you use the term 'haline contraction', but is it the case that the ocean becomes more saline and contracts in your simulations? If not, then the term is misleading.

This term is changed now and is called 'the steric contribution' to sea level changes.

L 219: change to 'i.e., the difference between accumulation...'

Done.

L 220: 'for all the forcing scenarios'

Done.

L 222: instead of the vague 'in a high warming scenario', please refer specifically to the scenario has you have named it.

Done.

In sections 4.3 and 4.4, and figure 2 and 3, there is confusion. Is SMB the same as mass balance in the figure? In the text SMB is the difference between accumulation and ablation, but the figure reports accumulation, calving and runoff, and apparently not the SMB, nor the amount of ice at any given moment, which would also seem a useful metric.

We thank the reviewer for this good remark. The SMB is indeed the difference between accumulation and ablation. The surface runoff is equal to ice melt and liquid precipitation minus retention. We chose to use this term because it also gives an idea

about the magnitude of freshwater coming from the surface. The other freshwater comes from ice discharge and basal melting below ice shelves (only for Antarctica). The total volume of ice that is lost is not practical for our purposes since not all ice that is lost contributes to sea-level, which is still our principal variable to investigate.

As a suggestion, the tiles of sections 4.3 and 4.4 could mention also ocean currents, since these results are also prominent there.

This is a good suggestion and we changed the titles towards: 'The Greenland ice sheet and the AMOC' and 'The Antarctic ice sheet and AABW'.

L 238: punctuation is missing.

Punctuation is added.

L 259: since glacial isostatic processes are included, consider mentioning this when mentioning the model description. Are these processes carried out by the is the land-surface module, which if I am not mistaken is part or ECBilt?

No, the glacial isostatic adjustment model is a component of the ice sheet models for Greenland and Antarctica. This is now clearly mentioned in the model description.

L 282: although an asymptotic behavior seems to emerge for all scenarios, it would be interesting to mention (and later discuss?) the late convergence between SLR of the 2.6 and 4.5 scenarios, which seems unexpected to me.

It is attempted to explain this by the slow melting of the Greenland ice sheet for scenario MMCP2.6. The overall ice sheet contribution is in the end very similar, part of the West Antarctic ice sheet retreats and the entire Greenland ice sheet.

L 291: after 10000 years of simulation, or after year 10000 of the simulation?

This has now been clarified by the statement: 'at 12,000 AD'.

L 298: what do you mean by 'inferred'? is it reconstructed/measured by use of/in proxies?

We mean 'reconstructed', and changed the terminology accordingly.

In the next sentence, I suggest adding mention of which two periods those combinations of sea level and CO2 concentrations refer to.

Done.

Next sentence still: I suggest mentioning here Figure 6.

Done.

On figure 6: it is bizarre that it does not show the -120m sea level for 180 ppm mentioned in the text, I guess because the figure only uses Foster and Rohling 2013

and that work did not include such low stand. Nevertheless,because that extreme is irrelevant to the range of values here, it seems acceptable.

Further in fig. 6: what is the vertical line, pre-industrial concentration?

While mentioned in the text, there is no red line for the linear fit in figure 6.

The vertical line is pre-industrial $CO_2$ concentration. We added the red line as the best fit for the data.

L 315: eliminate 'both of'.

Done.

In this context, a test to assess the likelihood that the data from this study belong to the distribution of data from the geological records would seem informative.

We consider a likelihood assessment here out of the scope of our comparison. Moreover, the comparison between the geological data and the data from this study clearly shows a similar trend.

L 318: thermal expansion seems quite flat for all scenarios, 10000 years into the simulations, so this can't plausibly be a contribution to further sea level rise that's meaningful for the scale of fig. 6.

The reviewer is right here and we excluded the last argument.

Discussion

L 321: this does not seem a suitable reference for future sea level rise. Many good references for this have already been cited in the introduction. Further, it's puzzling to see the discussion open with a contributor to SLR that is not the most relevant at present and by far not in the time-scales of this study.

We gave a better reference regarding sea-level rise during the 21st century due to glacier melting. The logic that is followed is to start with the components having a large contribution to sea-level at present (glaciers and ice caps and the steric contribution) and to discuss the components with a larger uncertainty (the ice sheets) afterwards.

L 322: are RCPs more appropriate than MMCPs here?

We wanted to use the introduced MMCP naming here to make the distinction between the numbers from our study (MMCP) and those from other studies that simulate changes up to the end of the 21st century (RCP).

L 330: are these numbers on the steric contribution from this study? Please clarify.

Yes, this is clarified.

L 333: what do you mean by 'updated', is this a different generation of climate models?

The term 'updated' refers to the use of coupled models, while the former numbers don't include the influence of ocean circulation changes on the steric contribution. On the other hand, the study of Hieronymus (2019) also doesn't include the coupling with ice sheets.

L 335: verb tense is wrong.

This is corrected.

L 336-337: it should be stated more clearly that the models included in the reference cited do not have coupling between ocean and ice-sheets (if that is the case).

Done.

L 339: 'local annual mean temperature' and 'mean SAT' seem to mean the same thing here, but different terminology is confusing.

We changed 'local annual mean temperature' to 'mean SAT'.

L 389: it is not clear whether the impacts on AABW and sea ice formation are from this study or from the references listed.

The impact on AABW and sea-ice formation are from the references listed.

Conclusion

L 406: Related to one of the main points above, it seems inappropriate to state that SLR is irreversible. That is, from your scenarios and results it appears irreversible absent active anthropogenic carbon sequestration, i.e., under the debatable assumption of no anthropogenic alteration of the carbon cycle beyond the atmospheric emission of CO2 and methane.

We brought more nuance to the formulation.

L 408: to 'simulate' 'in the real world' seems an oxymoron.

We removed 'in the real world'.

L 431: change to 'or the Antarctic ice sheet.'

Done.

Fig. 1, unlike other figures, has the two additional scenarios in dashed lines instead of solid lines. Whereas the reason is given in the caption, this lack of consistency across figures and panels is not advisable.

This is adapted and changed to solid lines.

Fig. 5a lacks the legend for the scenarios, which is all the more confusing because colors in 5b are used to another purpose.

A legend is added for Figure 5a.

Also, the caption may be confusing: is this GMSL due to all relevant processes, or is it necessary to list them all here making the reader think that maybe some other process is left out?

We changed the caption to 'GMSL rise during the next 10,000 years', to avoid confusion.

Finally, I am not sure that panel 5b is the most efficient way to show the timing difference between (cumulative) emissions and sea level change. A plot of those two quantities against time would have several benefits compared to 5b: it would show the timing aspect more clearly, it would show the scenarios, and would be much easier to read. Please take this as a suggestion.

We thank the reviewer for the suggestion. A plot of GMSL rise against time, showing the different forcing scenarios is already given in Figure 5a and we believe Figure 5b is a valuable addition by showing the sea-level rise in terms of cumulate $CO_2$ emissions for different time snapshots.

---

## Author Comment (AC3) · 21 Jul 2020

**Response to Anonymous Referee #2**

In this paper, the authors aim to investigate the evolution of Greenland, Antarctica ice sheets and minor components of sea-level rise over a long period (10ky) using an integrative strategy. A model of intermediate complexity, LOVECLIMv1.3, is coupled with ice sheets models (Greenland and Antarctica) that enable them to test different pCO2 and methane scenarios for the period of 1000 to 10000 years.They investigate indeed the response of the Earth climate system to a large but short lasting perturbation. It is necessary to run long simulations that account for long time response of deep-ocean, ice sheets and CO2 evolution.The authors first describe the tool they used, the originality of which is to account for feedback between atmosphere/ocean and ice sheet, then they describe the scenarios they chose and they finally present their results in terms of different contributions.The paper is well written and the issues are interesting, nevertheless there is room for improvements on several points.

**Author's response:** Thank you very much for the positive evaluation and useful comments.

1. Discussion of the limitation of the study and its possible consequences: The scenarios are prescribed from an initial perturbation based on the four RCPs of IPCC scenarios and 2 supplementary scenarios. This paper represents an improvement compared to previous studies because feedbacks between climate and cryosphere are accounted for. The discussion on the CO2 evolution,which is driven by different anthropic pathways,is only discussed from a "mathematical" point of view. The authors should discuss the limitation of such an approach. Indeed there are also interactions between carbon cycle and ocean and interactions between vegetation and carbon cycle, which are not limited to permafrost and clathrate destabilizations, and which are not accounted for. For instance, a dynamical vegetation model could be useful to account for the effect of desertification (albedo and water cycle).

**Author's response:** The reviewer is right that we do not discuss all possible feedbacks when using the impulse response functions. There are certain aspects such as the influence of a vegetation model on albedo and water cycle that we neglected and we emphasised this in the scenario description. On the other hand, by using the impulse response functions based on a literature study, we take into account the existing model uncertainty about the magnitude of the ocean uptake of CO2 and the uptake of CO2 by vegetation, since the impulse response functions are based on carbon cycle models.

Another issue that is not discussed is the long-term climate evolution. Indeed all scenarios depict a complete melting of the GRIS. The simulated climates are similar to Pliocene climate and thus the impact of orbital forcing/orbital parameters may be drastically modified in comparison with Quaternary large glacial/interglacial oscillations.

**Author's response:** We discussed the long-term climate evolution in terms of global, Greenland and Antarctic mean temperature anomalies. The discussion of long-term climate evolution is now extended by showing plots of the spatial temperature anomaly

above Greenland and Antarctica. Also, the influence on climate of the insolation changes is discussed in more detail and an additional figure showing the insolation changes at 70 ˚N and 70 ˚S for the period -130 kyr (onset of the Last Interglacial) to +50 kyr (next large boreal summer insolation minimum) is added, to put the insolation changes during the next 10,000 years into perspective (they are small on account of a low eccentricity).

2. Computation of grounding line evolution with coarse-grid modeling:The response of the grounding line is very important and should be discussed in more details because it is difficult to compute it using climate change simulation by a coarse-grid model. For instance, to test the capability of their model in order to compute correctly changes in grounding line, the authors could use the last deglaciation and they should validate their model over such a period.

**Author's response:** It is a valid remark from the reviewer to state that we use a rather coarse resolution Antarctic ice sheet model in our simulations. However, a similar model version with a resolution of 20 km has proven to be able to simulate the changes in the grounding line position for the Last Interglacial (Goelzer et al., 2016a) and the Last Glacial Maximum (Huybrechts, 2002). Also, we do a proper climatic spin-up over the period 1500-2000 AD, while the Antarctic ice sheet is spun up over the last four glacial cycles to carry the long-term ice sheet history. We believe that a full validation to simulate the last deglaciation is a study on itself and is far beyond the scope of this study. Moreover, the last deglaciation is not the best analogue for future ice sheet melting since it is a colder period with massive ice sheet melting from the Laurentide, Cordilleran and Fennoscandian ice sheets (ice sheets that are not included in our model set-up). A better analogy can be made with the Last Interglacial where the Greenland and Antarctic ice sheets were also reduced in size and where our model has proven to be valuable (Goelzer et al., 2016a, Goelzer et al., 2016b).

3. Parametrization and scenarios: Line 114-115: in addition to the fact that the authors wrote P11 in the text and P71 in the Table S3, this part of the paper is very unclear to me. The authors chose one set of parameters, and with this set, they provide 6 different simulations. All these scenarios do not need to be run again. But for the 2 extreme ones (RCP2.6 and feedback), we would like to have the result when using different parametrizations as far as LOVECLIMv1.3 needs parameterization to compensate the approximation made. Indeed we would like to know how much the results are dependent on the parametrization.

**Author's response:** We thank the reviewer to notice the mistake about the parameter naming. It should be P11 in Table S3 and it is now corrected. The different parameterizations mostly differ in the treatment of the long-wave radiation scheme (parameters amplw, explw, ampanir, ampanir2; see Supplementary Information). We did sensitivity tests with 2 other parameter sets: P32a (which is very similar to P32b) and P11 for the two extreme scenarios, as suggested by the reviewer. We now include a new paragraph in the result section that deals with the model uncertainty of the future sea-level change projections for scenario MMCP2.6 and MMCP-feedback.

Moreover, the scenario using methane from clathrate emission is important not only in terms of quantity of greenhouse gases emitted but these emissions

could last several kyears, which is the duration of Paleocene-Eocene thermal maximum. The authors should explain the reason of their choice.

**Author's response:** The reason of our choice is now explained in more detail in the scenario description and the rate of methane release from clathrate emission during the next millennia is now compared to the order of magnitude of methane emissions during the PETM.

"*MMCP-feedback assumes a moderate methane release from methane hydrates of 600 GtC by adding constantly $CO_2$ after 2250 AD (from the peak concentration onwards) until the end of the simulations (equivalent to a release of 6.15 GtC per 100 year), in accordance with the experiments of Archer et al (2009a). It is thought that methane hydrate dissolution caused a strong increase in atmospheric $CO_2$ levels with a possible total release of > 5000 GtC during the Paleocene-Eocene Thermal Maximum (PETM; Dickens, 2011). This release would have been caused by an initial warming trigger and might have lasted for more than 100 kyr (Zeebe and Lourens, 2019). Therefore, our estimate of the strength of the methane emission feedback is in the same order of magnitude as the methane emission rate during the PETM (5 GtC per 100 year). Also, it is thought that it takes a long time (> 10,000 years) before a significant part of the sediment has warmed in reponse to the ocean bottom temperature increase (Zeebe, 2013b), supporting our conservative methane release in comparison to the size of the methane reservoir.*"

If the authors account for these main comments and more analytical comments below, I consider that this paper is a valuable contribution to an important issue and should be published in Earth System Dynamics.

More detailed comments:

-Title: the authors use the term "semi-equilibrated",which is never clearly defined. I would prefer "quasi-equilibrium" but anyway the author should give an objective criteria for this term.

**Author's response:** We chose to use the term 'semi-equilibrium' because sea-level has almost adapted to the forcing after 10,000 years. Semi means 'half' or 'partly', which is a good description in our opinion of the sea-level changes after 10,000 years. The term 'quasi-equilibrium' is stronger because it means that the changes are infinitesimal small. This is a state that we didn't achieve at the end of the simulations.

-Abstract:

what is a semi-equilibrium?

**Author's response:** With a semi-equilibrium we mean that sea-level rise has almost equilibrated. In our experiments, we consider that a semi-equilibrium is reached when sea-level changes drop below 5 cm per century.

*"After 10,000 years, the sea-level change rate drops below 0.05 m per century and a semi-equilibrated state is reached"*

Over 10ka it may be important to account for astronomical forcing, especially for precession cycle (the period of interest here 1-10 ky is half the duration of precessioncycle).

**Author's response:** The astronomical forcing is included. We added this information to the abstract.

I don't really understand the last sentence of the abstract: there is no geologic analogue for the next 10ky in Earth history -as far as I know.

**Author's response:** We removed this last sentence from the abstract, since it is hard to explain the comparison with the geological archive in one sentence.

How is it possible to reach more than 5800 GtC?

**Author's response:** RCP8.5 (and its extension to 2300 AD) is equivalent to a cumulative emission of ~5300 GtC (270 GtC before 2000 AD and 5018 GtC between 2000 and 2300 AD; see Meinshausen et al., 2011). The inclusion of the methane feedback release would add another 600 GtC during the next 10,000 years.

1. Introduction

The introduction is fine but 2 topics should be introduced or developed:

1. The last deglaciation,which lasts around 10000 y,is an interesting period to validate the model used here. The authors should discuss this point,which is completely absent in the introduction.

**Author's response:** As stated before, we believe that a validation over the last deglaciation is a completely different scope, and moreover is not a particularly good analogue as it involved big ice sheets on the continents of the northern hemisphere which are no longer present. Therefore, we do not see the need to introduce this topic in the introduction and we would like to keep the focus on future sea-level rise and interactions with the Earth system.

2. The methane hypothesis should be clearly explained. There is a first short term feedback linked to the permafrost melting and a long term effect on clathrate destabilization. Concerning this second point, there are several unknowns: the quantity of methane, which is discussed by the authors but also the onset and the duration of these emissions,which they should discuss more.

**Author's response:** We thank the reviewer for this good suggestion and added a discussion about the methane hypothesis in the introduction. We further extended the explanations in the scenario description section.

*"Several feedbacks in the climate system reinforce an initial perturbation. The increase in polar temperatures releases methane from permafrost regions. This process is slow*

*and accelerates climate change on a centennial timescale (Schuur et al., 2015). It is believed that a warming ocean could potentially also release a massive amount of methane from methane clathrates, somehow similar to what has happened at the Paleocene-Eocene Thermal Maximum (PETM; Zachos and Zeebe, 2013). The PETM took place at about 56 Myr and was characterized by a rapid global temperature rise of more than 5 °C in addition to a strong background warming, probably caused by the massive release of methane from clathrate hydrates (Dickens, 2011). It is thought to be the best analogue for future climate warming in a strong greenhouse world (Zeebe and Zachos, 2013)."*

2. Model description and initialization

The authors should clarify how they downscale the large grid of atmospheric and ocean models to high resolution ice sheet models (GRIS or AIS).

**Author's response:** We explain the use of anomaly forcing to force ice sheet models and the use of a PDD model to obtain melt rates on the high resolution grids of the ice sheet models. No other downscaling techniques are included.

Line 105: what are the range of corrected biases for present day climate?

**Author's response:** The mean error for the present day climate is 0.38 °C over Antarctica (range from -1.2 °C to + 0.9 °C) and 0.44°C over Greenland (range of -1 °C to 1.5°C). We added this information to the model description.

Line 106: is the PDD really appropriate for this study? Why did the authors use a method based on present day (cold context) rather than a method based on energy balance?

**Author's response:** The PDD model is a computationally efficient method to calculate the mass balance and allows to take into account the melt on the much finer ice sheet grid. Above all, the method takes into account the important feedback of height changes on mass balance. An energy balance model introduces more, in part poorly known, parameters and is therefore not necessarily superior.

Concerning the choice of parameters (Table S3) the authors should justify this choice and its possible consequences.

**Author's response:** We have justified our choice of the preferred parameter set P22 based on the sea-level change projections by 2300 AD and by the predicted polar temperatures by the end of the century. In parameter set P11, P32a and P32b, we had less confidence because of their worse performance during the coming decades to centuries in comparison with high resolution, coupled ice sheet-climate modelling studies.

Line 117: the tuning of parameterson the last 500y is not really appropriate to explore large changes of cryosphere. The last deglaciation is certainly a better but more complex target for the goal of this paper. Moreover, the authors selected a

parameterization "because of its mid-range contribution to sea-level at 2100 AD and 2300 AD in comparison with recent studies". Is it a correct criteria in science to be "mid-range"? I think the authors should favor more physically based parametrizations. We also would like them to use other parameterizations for the scenarios including feedbacks.

**Author's response:** The different parameter sets were used to explore the climate and sea-level evolution over the next millennia. P22 is a parameter set that has been used in previous studies (Goosse et al., 2007; Loutre et al., 2011, Loutre et al., 2014, Goelzer et al., 2016a, Goelzer et al., 2016b) focusing on the last millennium or the Last Interglacial. The other parameter sets had a lower (P11) or higher (P32a and P32b) climate sensitivity and differed mostly in the longwave radiation scheme. As explained before, we had little trust in these parameter combinations on the longer term and therefore chose P22 as the preferred parameter set. Nevertheless, we added now sensitivity experiments for climatic model parameter set P11 and P32a using scenario MMCP2.6 and MMCP-feedback, as suggested by the reviewer, to explore the climate model uncertainty.

3. Scenario description

Line 150-153: scenario RCP8.5 leads to a maximum PCO2 of around 5000 ppm about 20 PAL. But is there enough fossil fuel to be burnt to achieve such a value?

**Author's response:** The maximum pCO2 is around 2000 ppm and the total emissions are around 5000 GtC. The uncertainty in fossil fuel reserves is so large that we decided not to base our experiments on speculative estimates but chose to extend the IPCC RCP scenarios.

4. Climate response and global sea-level budget of individual terms

The last deglaciation is strongly nonlinear, with acceleration, as during meltwater pulse and reduced SLR during colder episodes.The ability of the model used here to reproduce the deglaciation should be discussed.

**Author's response:** As discussed before, we believe that the last deglaciation is not the best analogue for future ice sheet melting. Meltwater pulses were related to fast reductions in the Laurentide or Eurasian ice sheets which are no longer there, or perhaps from fast ungrounding of the Antarctic ice sheet from its glacial maximum extent, different from the current geometry of the Antarctic ice sheet.

A plot showing both terms accumulation and ablation in the different scenarios could be interesting to be depict and discuss.

**Author's response:** We show plots of accumulation and runoff (ablation and liquid precipitation), which is nearly the same.

For GRIS what is the ocean dynamics in North Atlantic? It seems that the AMOC recovers and is even stronger than for PD: does that mean that the GRIS could not be covered by perennial ice sheet for long periods?

**Author's response:** We extended the discussion about the AMOC and added references about its behaviour and the past (mid-Pliocene) and future. Our results are in line with the simulations of Jansen et al. (2018), who suggested that the AMOC strength increases in response to surface warming.

5. Long-term sea level rise in the light of the geological record

Line 312 and Fig 6
This is an interesting comparison. Nevertheless,there should be an hysteresis between the values of melting GRIS and AIS in the simulations described here and the pCO2 values corresponding to the onset of the same ice sheets during Cenozoic due to the changes of surface albedos.

**Author's response:** We thank the reviewer for this good remark. We improved the discussion by adding information on hysteresis and the difference between the onset of ice sheet growth versus ice sheet decay.

*"The geological estimates of sea-level high stands includes data that are affected by the hysteresis effect between ice sheet growth and ice sheet decay, where an ice sheet can either exist or not exist at a certain $CO_2$ level depending on its history (Pollard and DeConto, 2005). Also the effect of a different paleotopography might have had an influence on the inception of ice on paleotimescales where the AIS could grow for lower $CO_2$ values during the Oligocene than at present (Paxman et al., 2019). Both arguments suggest that our curve (red line) is expected to be below the best estimate of the geological data (blue line), requiring higher $CO_2$ levels to melt the ice sheets on Earth in the future than during periods in geological history."*

6. Discussion

Line 360:There are two sources of uncertainties that the authors should comment with more details:

1. The AMOC evolution because in PLIOMIP2, most of the models depicted a lower AMOC,which provided a context not favorable to the onset of GRIS

**Author's response:** We added information to the discussion about the AMOC response during the mid-Pliocene from PLIOMIP2 studies and about the simulated future evolution.

2. The authors should compare their highest scenarios to the resources available in terms of fossil fuel.The authors pointed out the possible mechanisms that speed up the AIS melting as Marine Ice Cliff Instability MICI. Moreover,the record of the previous deglaciation show much variability in SLR rise with acceleration and regression of the ice sheets. These simulations did not reproduce this variability in future scenarios maybe partly due to coarse resolutions.There are anyway hot debates on the prevision of SLR only concerning the end of this century and very few models are able to reproduce

using transient experiments the SLR during the last glacial/interglacial cycle.For all these reasons, the authors should be careful of uncertainties and limitations of their strategy even if it is a consistent approach for investigating long time scale.

**Author's response:** We have added the model uncertainty for the extreme forcing scenario. For scenario MMCP-feedback, melting of all land ice on Earth cannot be excluded, given the model uncertainty. In this scenario, there is a strong acceleration in sea-level rise after 7000 model years due to the strong albedo-temperature feedback that initiates once the ice sheet retreats on land, somehow similar to the strong increase in melt at the last deglaciation when the Laurentide ice sheet disappeared. We included an extra paragraph on these results.

Moreover, the evolution of the grounding line when using coarse grid model is difficult to capture and therefore, it is another limitation of the paper that the authors should comment.

**Author's response:** We believe that we gave a sufficient discussion of the model limitations and performance on L362-371 in the Discussion.

**7. Conclusion**

Their main results are summarized: A GRIS melting for all scenarios, this melting being irreversible in the time window of interest here whereas the AIS contributes to the SLR differently, depending on scenarios.These results are consistent with a GrIS becoming perennial only when CO2 is around 300-400 ppm whereas AIS needs a higher CO2 level to become perennial around 800 ppm.The authors should also discuss the consistency of their results when compared to the evolution of these ice sheets.

**Author's response:** It is not clear to us what the reviewer intends to say, nor how we should respond to it.

Comments on Figures

General comment for all the Fig. : In fact, while GMSL rises to high values (env. 30 m) many regions that are considered in the simulation as land points shift to ocean points. By the way, the ocean/continent distribution is changing with SLR. This is certainly a minor effect for the coarse grid model for several meters. But it may beimportant when SLR reaches 10 to 30 meters. This effect is not accounted for in the figure and I believe also that this distribution is kept fixed in all simulations. Thus, the authors should discuss this approximation.

**Author's response:** It is indeed true that we neglect the effect of GMSL changes on the plots of the GrIS and the AIS and we clearly mentioned this approximation in the given figures.

Fig 2: it could be interesting to show also the GRIS configuration for present day.Fig 3: interesting to have AIS for present day. Moreover,it could be interesting to add a snapshot at 3ky with WAIS and EAIS separately.

**Author's response:** We have added a figure with the configuration of the Greenland and Antarctic ice sheet simulated at the start of the simulations. We did not distinguish between the East Antarctic ice sheet and the West Antarctic ice sheet because this would require to define the ocean basins beforehand, which we did not do in this study.

Figure 7: the logic of the caption (a) (b) succession is not easy to follow at first sight.

**Author's response:** We adapted the figure caption to increase the logic.